# Molecular basis of synaptic specificity by immunoglobulin superfamily receptors in *Drosophila*

**Shouqiang Cheng[1], James Ashley[2], Justyna D Kurleto[1,3], Meike Lobb-Rabe[2,4], Yeonhee Jenny Park[1], Robert A Carrillo[2], Engin Özkan[1]***

[1]Department of Biochemistry and Molecular Biology, University of Chicago, Chicago, United States; [2]Department of Molecular Genetics and Cell Biology, University of Chicago, Chicago, United States; [3]Faculty of Biochemistry, Biophysics and Biotechnology, Jagiellonian University, Krakow, Poland; [4]Graduate Program in Cell and Molecular Biology, University of Chicago, Chicago, United States

**Abstract** In stereotyped neuronal networks, synaptic connectivity is dictated by cell surface proteins, which assign unique identities to neurons, and physically mediate axon guidance and synapse targeting. We recently identified two groups of immunoglobulin superfamily proteins in *Drosophila*, Dprs and DIPs, as strong candidates for synapse targeting functions. Here, we uncover the molecular basis of specificity in Dpr–DIP mediated cellular adhesions and neuronal connectivity. First, we present five crystal structures of Dpr–DIP and DIP–DIP complexes, highlighting the evolutionary and structural origins of diversification in Dpr and DIP proteins and their interactions. We further show that structures can be used to rationally engineer receptors with novel specificities or modified affinities, which can be used to study specific circuits that require Dpr–DIP interactions to help establish connectivity. We investigate one pair, engineered Dpr10 and DIP-$\alpha$, for function in the neuromuscular circuit in flies, and reveal roles for homophilic and heterophilic binding in wiring.
DOI: https://doi.org/10.7554/eLife.41028.001

***For correspondence:**
eozkan@uchicago.edu

**Competing interests:** The authors declare that no competing interests exist.

## Introduction

Maps of synaptic connectivity establish robust neuronal networks defining circuit function and behavior. Modified local or global connectivity is observed in numerous neurodevelopmental disorders, including schizophrenia and autism (*Calhoun et al., 2011*; *Khan et al., 2013*; *Supekar et al., 2013*). Furthermore, genes that govern wiring processes are commonly associated with such diseases (*Mitchell, 2011*). However, establishment of the proper connectivity is not a trivial process: There appears to be no correlation between how widely two neurons contact each other and how often they would participate in a synapse (*Kasthuri et al., 2015*), and the large numbers of neuronal cell types in dense neuropils present a significant challenge for how a genetically encoded program can establish the specific connectivity patterns.

In stereotyped neuronal networks, synaptic connectivity is believed to be dictated by cell surface proteins and secreted molecules, which can (1) assign unique identities to neurons, and (2) mediate axon guidance and synaptic targeting functions through specific interactions with their cognate ligands and receptors. This paradigm of *chemoaffinity*, first elaborated by Roger Sperry, has been supported by the discovery of a number of molecules that participate in wiring-related functions, especially in axon guidance (*Sanes and Zipursky, 2010*). Many of these chemoaffinity molecules function as neuronal adhesion molecules, and are conserved across animal taxa from nematodes to

mammals. Nevertheless, discovery of synapse-targeting adhesion molecules, proteins that determine which pairs of neurons will create synapses, has been limited.

We have recently identified interactions between two groups of *Drosophila* IgSF (immunoglobulin superfamily) cell adhesion molecules with properties closely matching a neuronal chemoaffinity function: 21 Dpr proteins, named after the founding member Defective proboscis extension response (*Nakamura et al., 2002*), selectively bind nine proteins, now called the Dpr-interacting proteins, or DIPs (*Özkan et al., 2013*). Dprs and DIPs are expressed across the nervous system. In agreement with the paradigm that they can serve as 'identity tags' on neurons, different combinations of Dprs and DIPs are known to be expressed on different neuronal classes in the optic lobe, giving neuronal surfaces unique 'identity codes' (*Carrillo et al., 2015*; *Tan et al., 2015*). A similar expression pattern has been observed in the ventral nerve cord (*Özkan et al., 2013*) and the olfactory system (*Barish et al., 2018*). Most importantly, in the knockouts of the interacting pair Dpr11 and DIP-γ, synapse targeting defects have been observed in the optic lobe for synapses formed between Dpr11 and DIP-γ-expressing neurons (*Carrillo et al., 2015*). Therefore, Dprs and DIPs are strong candidates for a synapse specification or targeting function. In addition, they have been shown to be necessary for neuronal survival in the optic lobe, and synapse maturation of neuromuscular junctions (*Carrillo et al., 2015*), both important aspects of establishing functional neural circuits.

To mediate a wiring specificity function, Dprs and DIPs cannot promiscuously interact with all possible binding partners. Accordingly, out of 189 possible Dpr–DIP interaction pairs, only 36 Dpr–DIP interactions could be demonstrated (*Carrillo et al., 2015*). The mechanisms by which molecular recognition, and therefore cellular connectivity, is established between cognate Dpr–DIP pairs is not clear: Our crystal structure of the first Dpr–DIP complex, Dpr6 bound to DIP-α, showed a role for shape complementarity, but no clear determinants of specificity were identified (*Carrillo et al., 2015*). Comparative structural studies are necessary for revealing how similar sets of Dpr and DIP molecular interfaces can be used to create a multitude of productive protein complexes, while excluding others.

In this study, we undertook a structural, biophysical and biological characterization of Dpr and DIP adhesive complexes to explain the molecular basis of Dpr–DIP specificity and function. We present several crystal structures, including the complexes of three Dprs for which a neural phenotype has been demonstrated: Dpr11 with DIP-γ, Dpr1 with DIP-η and Dpr10 with DIP-α. We compare the interaction interfaces of heterophilic and homophilic complexes with respect to differences that lead to specificity as well as interaction energetics. Furthermore, we investigate structure-based rational design and strategies for switching affinities between Dprs and DIPs. Specifically, we demonstrate that structure-based mutagenesis of selected Dpr–DIP pairs can be used to study specific wiring defects and are useful tools for understanding circuit wiring in the *Drosophila* nervous system. With these tools, we establish that homo- and heterodimeric interactions of DIP-α are both required for proper wiring between a motor neuron and a postsynaptic muscle target. Overall, the work presented here provides a biochemical and structural framework for investigating protein families that may function as molecular specificity molecules in synaptic partner matching.

## Results

### Sequence relationships of Dpr and DIP subclasses explain the Dpr–DIP interactome

All 21 Dprs and nine DIPs are predicted to contain two and three immunoglobulin (IG) domains, respectively (*Figure 1a*). Dpr and DIP sequences can be easily identified across all major arthropod groups; however, there is little to no sequence conservation outside the predicted immunoglobulin domains. Despite the lack of conservation, most Dprs and DIPs contain signal sequences at their N termini, and hydrophobic patches, likely to be transmembrane helices or GPI linkage sites, at or close to their C termini. Therefore, we predict Dprs and DIPs to be cell surface glycoproteins. Our previous work identified the first IG domains, termed IG1, as the domains mediating the Dpr–DIP interaction by the formation of a pseudo-symmetric IG1-IG1 heterocomplex using the *GFCC'C'* face of the immunoglobulin fold (*Carrillo et al., 2015*).

Dpr and DIP sequences covering the IG domains can be aligned within each of the families to create phylogenetic trees, which demonstrate that closely related Dprs interact with the same DIPs, and

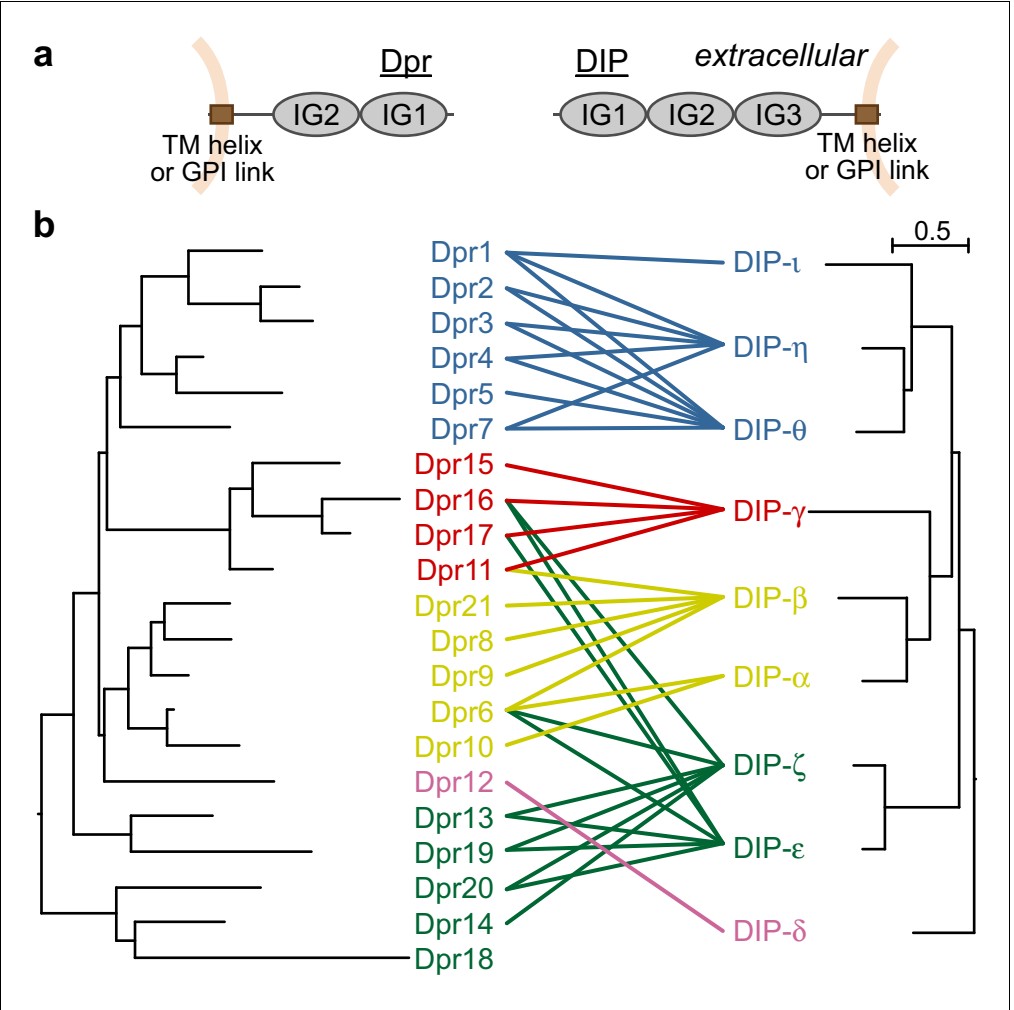

**Figure 1.** Dprs and DIPs can be classified into five classes based on sequence relationships and their interactions. (a) Cellular topologies and domain compositions of Dprs and DIPs. (b) Phylogenetic tree of Dprs and DIPs based on sequence alignments covering all IG domains. The colored lines indicate observed interactions mediated by IG1s. The scale bar represents 0.5 substitutions per site.

DOI: https://doi.org/10.7554/eLife.41028.002

The following figure supplement is available for figure 1:

**Figure supplement 1.** Statistical Analysis of artificial chimeras of cognate Dpr/DIP sequences.

DOI: https://doi.org/10.7554/eLife.41028.003

closely related DIPs interact with the same set of Dprs (*Özkan et al., 2013*) and *Figure 1b*): The average sequence identities in IG1 domains for Dprs and DIPs are 44 ± 9% and 53 ± 10%, respectively (±indicates standard deviation). The closely related DIPs-η and -θ, 71% identical in their IG1 domains, commonly interact with Dprs 1, 2 and 3, which are 65% identical in IG1. Based on phylogeny, Dprs and DIPs can both be classified into five subclasses, and each Dpr subclass can be assigned to a DIP subclass as cognates. 31 out of 36 interactions reported in *Carrillo et al. (2015)* are between the cognate Dpr and DIP subclasses. Therefore, the evolutionary histories of the Dprs and DIPs greatly explain the Dpr–DIP interaction network; however, a molecular and structural basis for specificity of Dpr–DIP interactions has remained elusive.

## Shared and divergent features in the structures of Dpr–DIP heterocomplexes

Our first structure, Dpr6 IG1 bound to DIP-α IG1 +2, showed highly complementary interaction surfaces (shape complementary value, $sc$ = 0.74), but included one hydrogen bond pair, two marginal hydrogen bonds (at 3.5 Å donor-to-acceptor distance) and no salt bridges between Dpr and DIP side chains, leaving shape complementarity as the major strategy for explaining Dpr6–DIP-α specificity (*Özkan et al., 2013*). Since there is significant sequence diversity for residues at the Dpr–DIP interface, it was not clear if this explanation would hold for the 35 remaining Dpr–DIP pairs.

For a comparative analysis of Dpr–DIP complexes, we set out to determine structures of complexes of Dprs and DIPs from branches distant to Dpr6 and DIP-α. We chose complexes of Dpr1 and Dpr11, as they are reported to have neuronal phenotypes (*Carrillo et al., 2015*; *Nakamura et al., 2002*), and therefore these complex structures can be directly used to investigate the relationship between Dpr–DIP interactions and the observed phenotypes. We crystallized and solved the structures of Dpr1 IG1 with DIP-η IG1, and Dpr11 IG1 with DIP-γ IG1 +2 (*Table 1*). Overall, the three IG1-IG1 heterodimers, including the Dpr6–DIP-α pair, can be confidently overlaid: the average root-mean-square deviation (RMSD) of all IG1 Cα atoms within the three complex structures is 0.78 Å (±0.14). All three complex interfaces are comprised of the same set of residues belonging to the *GFCC'C'* face of the IG domains (*Figure 2*). Interestingly, while the pairwise sequence identities among the three Dpr–DIP pairs are comparable (49% to 56%), the Dpr11–DIP-γ structure is significantly different than the other two. When the interface residues at the Dpr subunits are superimposed, the three DIP subunits are slightly displaced, with DIP-γ (dark gray in *Figure 3*) more distant from the other DIPs (~1.2 Å at the interface and up to 3 Å at the back face of the IG domain, *Figure 3a*; see *Figure 3—figure supplement 1b* for details of the displacement at the interface). Hence, different Dpr–DIP complexes can be established not only through shape complementarity between Dpr and DIP surfaces, but also by small but significant movements of the Dpr and DIP monomers with respect to each other, a mechanism not commonly recognized for related interaction pairs.

During our crystallization trials, we also grew crystals and determined the structure of DIP-γ IG1 +2 in a monomeric state. This has allowed us to compare three structures containing DIP IG1 and IG2 domains and the relative orientations of these IG domains (*Figure 2a*). In all structures, the two IG domains are in an extended conformation. This is unlike many multi-IG domain cell adhesion molecules known to adapt horseshoe-like structures, which require minimally four-amino acid linkers for the U turn (*Freigang et al., 2000*; *Sawaya et al., 2008*; *Su et al., 1998*), but is similar to cadherins (*Shapiro and Weis, 2009*) and certain classes of immunoglobulin superfamily receptors, such as the Synaptogenesis (SYG) proteins (*Özkan et al., 2014*). The extended conformation is due to lack of linker sequences between the two IG domains in DIPs, which also holds true at the predicted DIP IG2-IG3 and the Dpr IG1-IG2 boundaries. However, despite the lack of sizeable linker sequences, the DIP IG1-IG2 domain boundary is flexible, allowing for movement of the DIP IG2 with respect to IG1. This is a result of the lack of stabilizing influences such as calcium ions found in cadherin domain boundaries (*Shapiro and Weis, 2009*) or hydrogen bonds between the domains observed in SYG-1 and SYG-2 (*Özkan et al., 2014*). In the cases of cadherins and SYGs, rigidity of ectodomains was shown to be necessary for function and signaling. Lack of rigidity in Dprs and DIPs might indicate that they may not serve as signaling receptors directly and may not relay force or connect to cytoskeleton. This is corroborated by the fact that most Dprs and DIPs do not appear to have intracellular regions, supporting a model where Dprs and DIPs function as adhesion and specificity receptors on neuronal surfaces, but rely on co-receptors to relay signal intracellularly.

## Molecular details of Dpr–DIP complex interfaces driving specificity

We next compared the Dpr–DIP interaction surfaces of the three heterocomplexes of Dpr1, 6 and 11, which belong to different subclasses and should therefore present largest differences among heterocomplexes (*Figure 3b–d* and *Figure 3—figure supplement 1*). The centers of the interfaces are highly hydrophobic and conserved in sequence (marked in *Figure 3b* with *, *Figure 3c* and *Figure 3—figure supplement 1* (yellow side chains)), and likely provide significant energetic contributions to binding while not contributing to Dpr–DIP specificity. Yet, we also observed differences at these conserved positions at the structural level via rotameric changes and by rigid-body movements

**Table 1.** Data and refinement statistics for x-ray crystallography of Dpr–DIP-η and Dpr11–DIP-γ complexes, and DIP-γ alone.

| | Dpr1 IG1 + DIP-η IG1 | Dpr11 IG1 + DIP-γ IG1-IG2 | DIP-γ IG1-IG2 |
|---|---|---|---|
| **Data collection** | | | |
| Space Group | $P4_32_12$ | $P4_32_12$ | $P2_1$ |
| *Cell Dimensions* | | | |
| a, b, c (Å) | 74.08, 74.08, 235.45 | 85.36, 85.36, 103.58 | 29.33, 43.44, 86.14 |
| $\alpha, \beta, \gamma$ (°) | 90, 90, 90 | 90, 90, 90 | 90, 90.46, 90 |
| Resolution (Å) | 50–2.40 (2.44–2.40)[*] | 50–2.50 (2.65–2.50) | 50–1.85 (1.90–1.85) |
| $R_{sym}$ (%) | 14.0 (66.4) | 16.1 (181.8) | 14.6 (71.6) |
| $<I> / <\sigma(I)>$ | 22 (1.8) | 16.5 (1.2) | 8.8 (1.6) |
| $CC_{1/2}$ | (77.0)[†] | 99.8 (57.7) | 99.6 (82.8) |
| Completeness (%) | 93.4 (55.7) | 99.7 (98.1) | 99.9 (99.8) |
| Redundancy | 13.6 (4.7) | 13.3 (8.6) | 6.4 (3.8) |
| **Refinement** | | | |
| Resolution (Å) | 50–2.40 (2.49–2.40)[*] | 50–2.50 (2.69–2.50) | 50–1.85 (1.95–1.85) |
| Reflections | 25,031 | 13,767 | 18,695 |
| $R_{cryst}$ (%) | 21.22 (31.41) | 20.88 (29.94) | 20.45 (25.71) |
| $R_{free}$ (%)[‡] | 24.44 (34.15) | 26.29 (35.07) | 23.46 (30.55) |
| *Number of atoms* | | | |
| Protein | 3403 | 2312 | 1622 |
| Ligand/Glycans | 192 | 81 | 38 |
| Water | 84 | 69 | 179 |
| *Average B-factors (Å²)* | | | |
| All | 55.1 | 56.4 | 32.8 |
| Protein | 53.7 | 55.7 | 31.5 |
| Ligand/Glycans | 84.3 | 83.4 | 60.0 |
| Solvent | 46.3 | 47.3 | 38.6 |
| *R.m.s. deviations from ideality* | | | |
| Bond Lengths (Å) | 0.004 | 0.005 | 0.008 |
| Bond Angles (°) | 0.995 | 0.741 | 0.956 |
| *Ramachandran statistics* | | | |
| Favored (%) | 95.91 | 96.88 | 98.53 |
| Outliers (%) | 0.0 | 0.0 | 0.0 |
| Rotamer Outliers (%) | 0.0 | 0.0 | 0.0 |
| All-atom Clashscore [§] | 5.48 | 5.64 | 6.60 |
| Coordinate Error [¶] (Å) | 0.30 | 0.41 | 0.21 |

[*]The values in parentheses are for reflections in the highest resolution bin.

[†]Data processed by *HKL2000*, which does not report $CC_{1/2}$ for the entire resolution range of the data.

[‡]5% of reflections was not used during refinement for cross-validation: 1247, 707 and 933 reflections for the Dpr1–DIP-η, Dpr11–DIP-γ, and DIP-γ-only structures, respectively.

[§]As reported by *Molprobity*.

[¶] Maximum-likelihood estimate for coordinate error, reported by *phenix.refine*.

DOI: https://doi.org/10.7554/eLife.41028.007

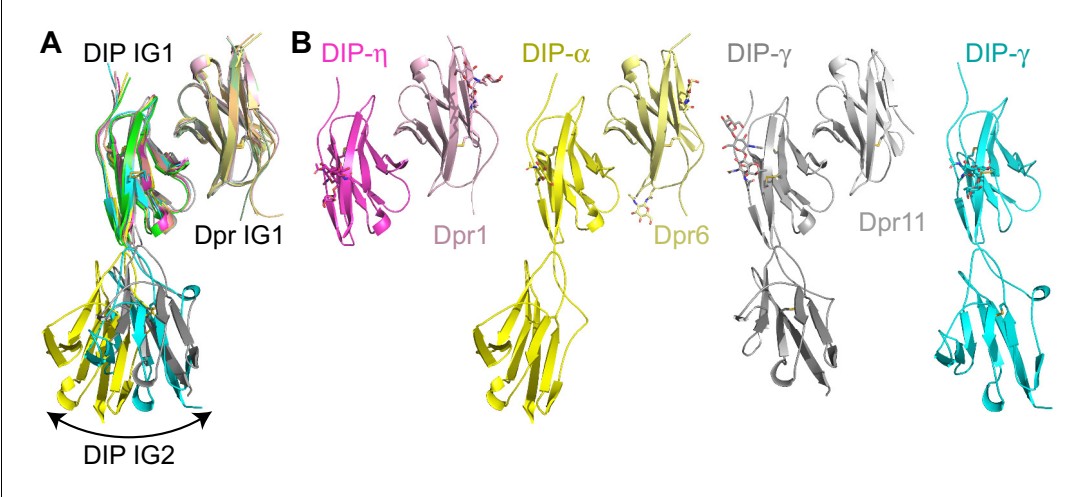

**Figure 2.** Structural comparison of three heterodimeric Dpr–DIP complexes. (a) Dpr1–DIP-η, Dpr6–DIP-α, and Dpr11–DIP-γ structures overlaid by aligning Dpr IG1 domains. (b) Side-by-side comparison of the structures.
DOI: https://doi.org/10.7554/eLife.41028.004

of DIPs with respect to Dprs (*Figure 3c* and *Figure 3—figure supplement 1c,d*), which allow for multiple complexes between Dprs and DIPs to form utilizing the same positions as conserved contact sites.

Outside the hydrophobic core, the interface residues are more polar, and significant differences in sequence and structure are present (*Figure 3d* and *Figure 3—figure supplement 1a* (cyan side chains)). To visualize how specificity is established, we focused on residue positions with stark differences among the three Dprs and DIPs in the periphery of the interface. Surprisingly, these sequence differences and their structural consequences, in several cases, cannot be explained by simple substitution of electrostatic or hydrophobic interactions. For example, Val164 in Dpr6 is a lysine in Dpr1 and Dpr11 (*Figure 3d$_1$*). However, this Val to Lys substitution is not accompanied by the presence of hydrophobic and acidic residues in DIPs directly interacting with this position. Instead, the presence of two Cγ atoms in Val164 results in crowding and an Ala82 in DIP-α, which is otherwise a valine in Dpr1 and Dpr11. Remarkably, the lysine residues replacing Val164 in Dpr1 and 11 do not form any salt bridges or hydrogen bonds to side chains in DIP-η and -γ, but serve in van der Waals interactions with DIPs.

A second highly variable position, Leu154 in Dpr11 (Lys in Dpr1 and His in Dpr6) also fails to explain specificity via simple electrostatic or hydrophobic matching (*Figure 3d$_2$*): The interfacing residue in the three DIPs is a glutamine or lysine (Gln78 of DIP-γ), which is pushed away by the hydrophobic Leu154 of Dpr1, but not in complexes with Dpr6 or 11. This movement of Gln78 in DIP-γ is then accommodated by a glycine in Dpr11 position 157, which is otherwise a bulkier and hydrophobic Leu and Ile in Dpr1 and 6. Therefore, the Dpr11–DIP-γ crystal structure enables us to see that Leu154 and Gly157 in Dpr11 are structurally linked and are needed for DIP-γ binding. Interestingly, the position equivalent to Gly157 of Dpr11 (and Dpr15) in the two other DIP-γ binders, Dprs 16 and 17, are larger but *polar* amino acids (*Figure 3d$_2$*), which can still be accommodated at the site as hydrogen bond participants with Q78 in DIP-γ.

We further looked to understand specificity via co-variation of Dpr and DIP residues in interacting pairs. We hypothesized that if there are sites in Dprs and DIPs that co-evolve, these could correspond to specificity determinants. For an analysis of sequence co-variation, we created artificial sequences where each sequence contained the IG1 from a Dpr, followed by the IG1 from a cognate DIP, resulting in 36 sequences. Covariation analysis by available tools is hindered due to the requirement for larger numbers of sequences. However, one method, the statistical coupling analysis (SCA) version 5 (*Lockless and Ranganathan, 1999*), identified one pair of amino acids, Dpr His94 (Dpr1) and DIP Met132 (DIP-η) (*Figure 1—figure supplement 1a*). These two residues directly contact each other and cap the hydrophobic interior of the interface (*Figure 1—figure supplement 1b*). In

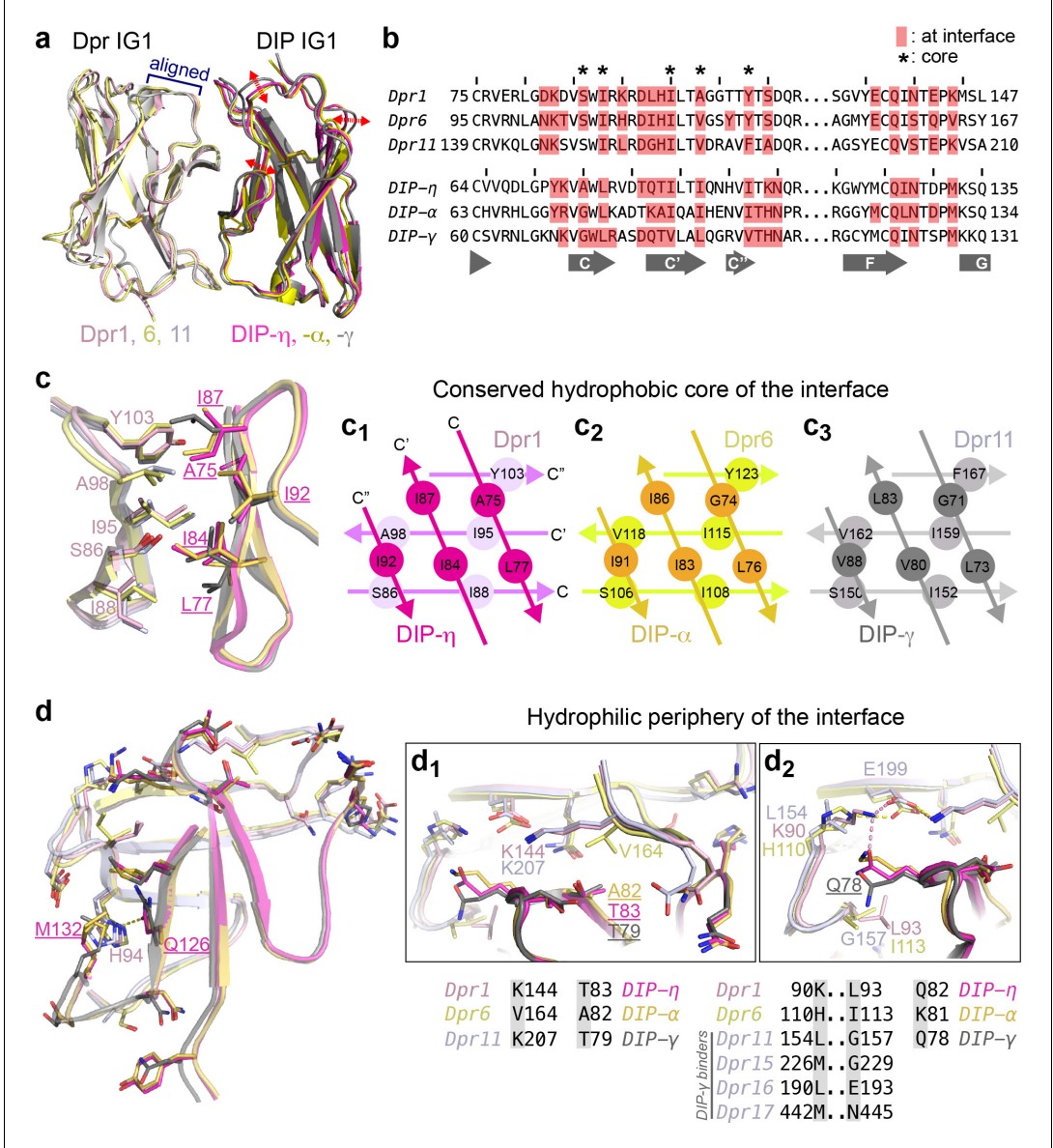

**Figure 3.** Conserved and variable features of interaction surfaces in the heterodimeric Dpr–DIP complexes. (a) Three heterophilic complexes overlaid by aligning the *GFCC'C'* sheets of the Dpr subunits. Displacement of the DIP subunits is illustrated with arrows. (Also see *Figure 3—figure supplement 1b*.) (b) Sequence alignment of parts of the IG1 domains from Dprs 1, 6, 11 and DIPs-η, α, and γ. Amino acids within 4 Å of the heterophilic partner, that is at van der Waals or hydrogen bonding distances at the Dpr–DIP interface, are labeled in red boxes. * indicates core interface positions in Dprs and DIPs. (c) The conserved hydrophobic core at the interface. The coloring scheme in *Figure 2* is used to distinguish Dprs and DIPs. Lighter colors present Dprs. Labels for DIP residues are underlined. Labels in light pink and magenta are for Dpr1 and DIP-η, respectively. Schematics in $c_1$ to $c_3$ show the conserved knob-and-hole interactions at the hydrophobic core. (d) The hydrophilic periphery of the interface. Labels for DIP residues are underlined. $d_1$ and $d_2$ show highly variable positions at the Dpr–DIP interface. For additional structural images, see *Figure 3—figure supplement 1*.

DOI: https://doi.org/10.7554/eLife.41028.005

The following figure supplement is available for figure 3:

**Figure supplement 1.** Conserved and variable features of interaction surfaces in the heterodimeric Dpr–DIP complexes.

DOI: https://doi.org/10.7554/eLife.41028.006

DIP-ε and -ζ, the methionine is replaced by an alanine, and ε-/ζ- binders Dprs 14, 18, 19 and 20 have non-histidine amino acids in the statistically coupled Dpr position (*Figure 1—figure supplement 1c*). It would be of future interest to determine the structures of DIP-ε and -ζ complexes to reveal the nature of the interaction at these positions. Overall, it appears that Dpr–DIP specificity is encoded not only by relationships between pairs of Dpr and DIP residues (e.g., K144 in Dpr1 with T83 in DIP-η, *Figure 3d₁*), but also through coupling of multiple residues, and through shape complementarity, where rotameric changes help create complementary surfaces.

## Energetics of the Dpr–DIP complex interface

While we could demonstrate and explain structural and amino acid differences between the three Dpr–DIP complexes through crystallography, static structures can rarely elucidate energetics of binding. To compare the three complexes from a thermodynamic point-of-view, we analyzed the same set of residues previously mutated in the Dpr6-DIP-α complex (*Carrillo et al., 2015*) in Dpr1–DIP-η and Dpr11–DIP-γ complexes via alanine mutagenesis, followed by heterophilic affinity measurements using surface plasmon resonance (SPR) (see *Figure 4a–d* for binding isotherms, and *Figure 4—figure supplement 1* to 3 for raw SPR data). The amino acids at the four positions in the three Dpr–DIP complexes (total of 24 positions) are shown in *Figure 4e*, and the effect of alanine mutagenesis, converted to ΔΔG values and fold-loss of binding, are in *Figure 4f* and *Figure 4—figure supplement 2*.

With these data, we first investigated the His94 (Dpr1) to Glu126 (DIP-η) hydrogen bond (*Figure 3d*), which appeared to be the only conserved side chain-to-side chain hydrogen bond among the three heterophilic complexes based on sequence conservation. For the Dpr1–DIP-η and Dpr6–DIP-α complexes, His-to-Ala mutation unexpectedly *increased* affinity despite removing a hydrogen bond and significant packing (*Figure 4a–f*). However, in Dpr11–DIP-γ, for which our structure unexpectedly shows no hydrogen bond, the His-to-Ala mutation abolished binding by more than five-fold. On the DIP side, the Gln-to-Ala mutation universally decreased or abolished affinity. These results indicate that even relatively conserved side chain-to-side chain hydrogen bonds can be dispensable for binding, and our ability to predict binding energetics based on static structures is limited. Interestingly, the His94 (Dpr1) residue is one of the statistically coupled residues mentioned above.

For hydrophobic side chains at the core of the interface, single-site alanine mutations consistently resulted in loss of affinity, and sometimes almost completely abolished binding (*Figure 4a–f*). We were not able to observe a rank order, or if a certain position is energetically more important across multiple complexes, that is a conserved hotspot. (*Figure 4—figure supplement 2a–b*). Therefore, we conclude that while the energetics of the interface shows some variation among the complexes, the hydrophobic conserved core of the interface provides much of the energy of binding, and the periphery is likely responsible for specificity.

## Structure-based alteration of dpr/DIP specificities

Engineered variants of Dprs and DIPs can be used to study wiring specificity in the *Drosophila* nervous system. In addition to the mutations described above, which decreased or increased affinities compared to wild-type, mutants with modified specificities can prove especially useful. For this purpose, we took a rational approach to modify Dpr11 to bind DIP-α. As DIP-α binds Dpr6, we substituted every interface amino acid in Dpr11 to its equivalent in Dpr6 (marked by * in *Figure 4g*), and performed a highly sensitive, high-throughput ELISA-like binding assay, the extracellular interactome assay (ECIA) (*Özkan et al., 2013*). This method can be used to report interactions with affinities as weak as approximately 1 mM (*Özkan et al., 2013*), and was used to originally discover Dpr–DIP interactions.

The first round of single-site and some double-site mutagenesis identified Dpr11 A165Y and two double mutants to weakly interact with DIP-α (marked by +, *Figure 4h*). Further installation of Dpr6 amino acids at the DIP-binding interface of Dpr11 slightly improved DIP-α affinity (*Figure 4i*). As a result of the second round of mutagenesis, we identified a triple-mutant Dpr11 variant, A165Y F167Y K207V (marked by *), which binds DIP-α and -γ. Interestingly, further non-exhaustive mutagenesis of the Dpr11 interface beyond A165Y F167Y K207V resulted in loss of binding to both DIPs. The identification of these residues is not accidental: two of the mutations are non-conservative changes in the polar periphery of the interface (*Figure 4—figure supplement 2c*); the K207 position

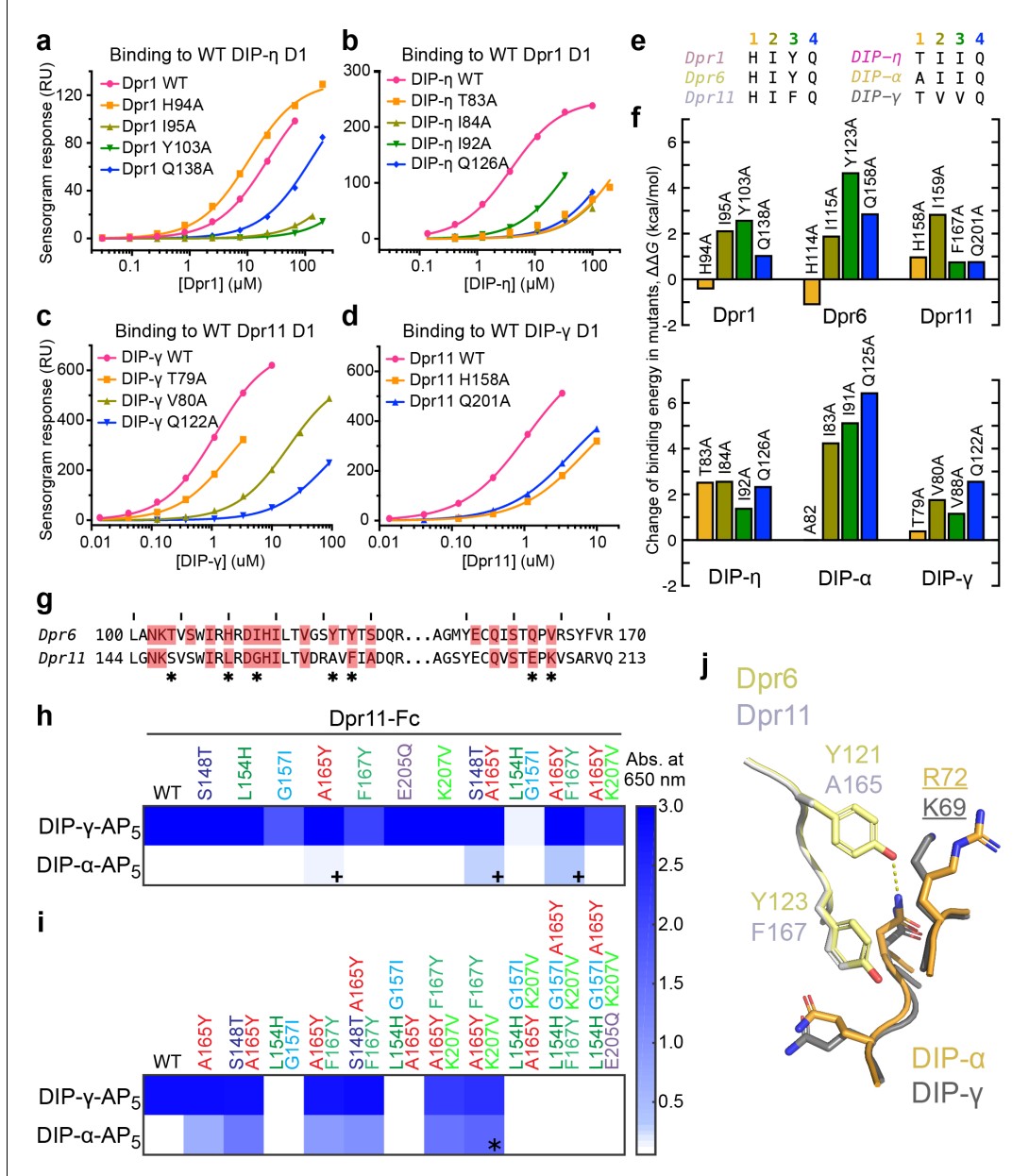

**Figure 4.** Energetics and engineering of Dpr–DIP interfaces. (**a–d**) Corresponding sets of Dpr and DIP interface residues are mutated in Dpr1 (**a**), DIP-η (**b**), Dpr11 (**c**) and *Carrillo et al., 2015*), DIP-γ (**d**) and *Carrillo et al., 2015*), Dpr6 and DIP-α (*Carrillo et al., 2015*). Binding isotherms for wild-type and mutants tested in this study are plotted with fits to a 1:1 interaction model. (**e**) The amino acids at the four mutated sites in six Dpr and DIP heterophilic partners. (**f**) Effects of alanine mutagenesis at the four sites in energy terms (from *Figure 4—figure supplement 2a*). (**g**) Comparison of Dpr6 and Dpr11 IG1 sequences. * highlights variable amino acids at the interface. (**h,i**) Binding of Dpr11 mutants to the native partner DIP-γ and the engineering target DIP-α using ECIA in two cycles. (**j**) Comparison of the interactions of conversion mutation sites (A165Y and F167Y) between the Dpr6–DIP-α and Dpr11–DIP-γ complexes.

DOI: https://doi.org/10.7554/eLife.41028.008

The following figure supplements are available for figure 4:

**Figure supplement 1.** SPR data for Dpr1 mutants binding to DIP-η.
DOI: https://doi.org/10.7554/eLife.41028.009

**Figure supplement 2.** Mapping of binding energetics onto structure.
DOI: https://doi.org/10.7554/eLife.41028.010

**Figure supplement 3.** SPR data for DIP-η mutants binding to Dpr1.
DOI: https://doi.org/10.7554/eLife.41028.011

*Figure 4 continued on next page*

*Figure 4 continued*

**Figure supplement 4.** SPR data for Dpr11 and DIP-γ WT and mutants.
DOI: https://doi.org/10.7554/eLife.41028.012

was already highlighted as a specificity determinant above, and in *Figure 3d₁*. The A165Y mutation is expected to create of a hydrogen bond absent in the Dpr11–DIP-γ complex but present in Dpr6–DIP-α and the engineered Dpr11 A165Y F167Y K207V–DIP-α complex (*Figure 4j*). Overall, this set of experiments demonstrates that Dpr/DIP specificities can be modified through rational design and the use of an inexpensive, high-throughput, fast and sensitive interaction assay.

## DIP homodimers are structurally similar to Dpr–DIP heterodimers

As we purified and crystallized several Dpr–DIP complexes, we were also able to grow crystals of DIP-η IG1 and determined its structure at 1.9 Å resolution, which revealed a homodimer (*Table 2*). These crystals only grew in the absence of Dpr1, which indicated that the heterodimers are likely more stable than the homodimer under the crystallization conditions used. The homodimeric DIP-η structure closely mimicked the Dpr1–DIP-η heterodimer: When DIP-η monomers were aligned, the other subunits, DIP-η in the homodimer and Dpr1 in the heterodimer, were only displaced by an RMSD of 0.78 Å for 87 out of 102 Cα atoms, excluding the variable DE loop and the mobile half of the A strand (*Figure 5a*). This is comparable to differences observed between heterophilic complexes. The main chain positions of a DIP-η bound to either another DIP-η or Dpr1 are virtually identical, and surprisingly, most side chains also preserve their rotameric states (*Figure 5b* and *Figure 5—figure supplement 1b*). On the other side of the interface, sequence differences between Dpr1 and DIP-η appear to not cause large deviations in the main chain atom positions between the two complexes (*Figure 5c*).

The crystal structures also reveal how DIP-η can accommodate binding to both Dpr1 and itself, with significant differences in sequence at the interface (*Figure 5—figure supplement 1a* and *Figure 5d*). For example, Tyr103 in Dpr1 (Y, F, or H in Dprs) is replaced by Ile92 in DIP-η (I or V in DIPs). This results in rotameric differences in close-by residue Ile87 in the common DIP-η subunit, which is further accommodated by other rotameric changes in the common DIP-η subunit, and sequence differences between the heterophilic and homophilic partners (*Figure 5d*).

The DIP-η homodimers observed in crystals also exist in solution. In size-exclusion chromatography experiments, elution volumes of DIP-η is dependent on the amount of protein loaded on the column (*Figure 5e*), which is a strong indication of homodimer formation with a dissociation constant on the order of protein concentrations used in the chromatography experiment, that is micromolar, and a monomer-dimer exchange rate much faster than the timescale of the experiment, which is also compatible with micromolar binding. While size-exclusion chromatography is not an equilibrium experiment, a binding isotherm based on elution peak volumes can be calculated (*Figure 5—figure supplement 1c*), yielding an apparent $K_D$ of 11–45 μM, an order of magnitude weaker than the heterodimer. Furthermore, we performed SPR experiments where low density DIP-η surfaces are created to prevent homodimers on chip surface, and data is fit to a binding isotherm while correcting for DIP-η dimerization in solution, yielding a $K_D$ of 14 μM (see *Figure 5—figure supplement 1d* and *Materials and methods* for details).

## Comparison and engineering of closely related Dpr–DIP heterophilic complexes

We next set out to compare heterophilic complexes with one common binding partner. We determined the crystal structure of Dpr10 IG1 bound to DIP-α IG1 and compared it to our Dpr6–DIP-α structure (*Figure 6*). The IG1 of Dpr6 and Dpr10 are 75% identical in sequence, and the interface only has three residues out of 19 that are different between Dpr6 and Dpr10 (*Figure 6a*). The two complex structures closely match each other (*Figure 6b* and *Figure 6—figure supplement 1a*), and unlike the differences among complexes described above, rotamers are nearly all conserved at the two interfaces. The differences in sequence are accommodated by small movements in surrounding side chains and do not propagate further (*Figure 6—figure supplement 1b*). Overall, these results

**Table 2.** Data and refinement statistics for x-ray crystallography of DIP-η–DIP-η and Dpr10–DIP-α complexes.

| | DIP-η IG1 + DIP-η IG1 | Dpr10 IG1 + DIP-α IG1 |
|---|---|---|
| **Data collection** | | |
| Space Group | C2 | P1 |
| *Cell Dimensions* | | |
| a, b, c (Å) | 88.43, 67.13, 61.01 | 51.01, 53.55, 56.69 |
| $\alpha, \beta, \gamma$ (°) | 90, 128.82, 90 | 119.68, 103.77, 92.88 |
| Resolution (Å) | 50–1.90 (1.94–1.90)* | 50–1.80 (1.91–1.80) |
| $R_{sym}$ (%) | 4.1 (55.9) | 3.3 (51.5) |
| $<I>/<\sigma(I)>$ | 12.3 (2.0) | 11.8 (1.3) |
| $CC_{1/2}$ | 99.8 (90.1) | 99.9 (67.0) |
| Completeness (%) | 98.3 (93.3) | 86.8 (53.8) |
| Redundancy | 3.4 (3.3) | 1.8 (1.7) |
| **Refinement** | | |
| Resolution (Å) | 50–1.90 (1.98–1.90)* | 50–1.80 (1.84–1.80) |
| Reflections | 21,783 | 40,105 |
| $R_{cryst}$ (%) | 23.21 (37.57) | 17.43 (39.51) |
| $R_{free}$ (%)† | 26.93 (45.88) | 20.54 (53.17) |
| *Number of atoms* | | |
| Protein | 1690 | 3421 |
| Ligand/Glycans | 40 | 282 |
| Water | 21 | 296 |
| *Average B-factors (Å²)* | | |
| All | 68.2 | 42.8 |
| Protein | 67.9 | 41.2 |
| Ligand/Glycans | 86.4 | 58.5 |
| Solvent | 55.2 | 46.1 |
| *R.m.s. deviations from ideality* | | |
| Bond Lengths (Å) | 0.003 | 0.008 |
| Bond Angles (°) | 0.636 | 0.938 |
| *Ramachandran statistics* | | |
| Favored (%) | 97.60 | 97.85 |
| Outliers (%) | 0.0 | 0.0 |
| Rotamer Outliers (%) | 1.04 | 0.53 |
| All-atom Clashscore ‡ | 1.44 | 3.59 |
| Coordinate Error § (Å) | 0.16 | 0.23 |

*The values in parentheses are for reflections in the highest resolution bin.

†5% of reflections was not used during refinement for cross validation: 1247 and 2002 reflections for the DIP-η–DIP-η and Dpr10–DIP-α structures, respectively.

‡As reported by *Molprobity*.

§Maximum-likelihood estimate for coordinate error, reported by *phenix.refine*.

DOI: https://doi.org/10.7554/eLife.41028.015

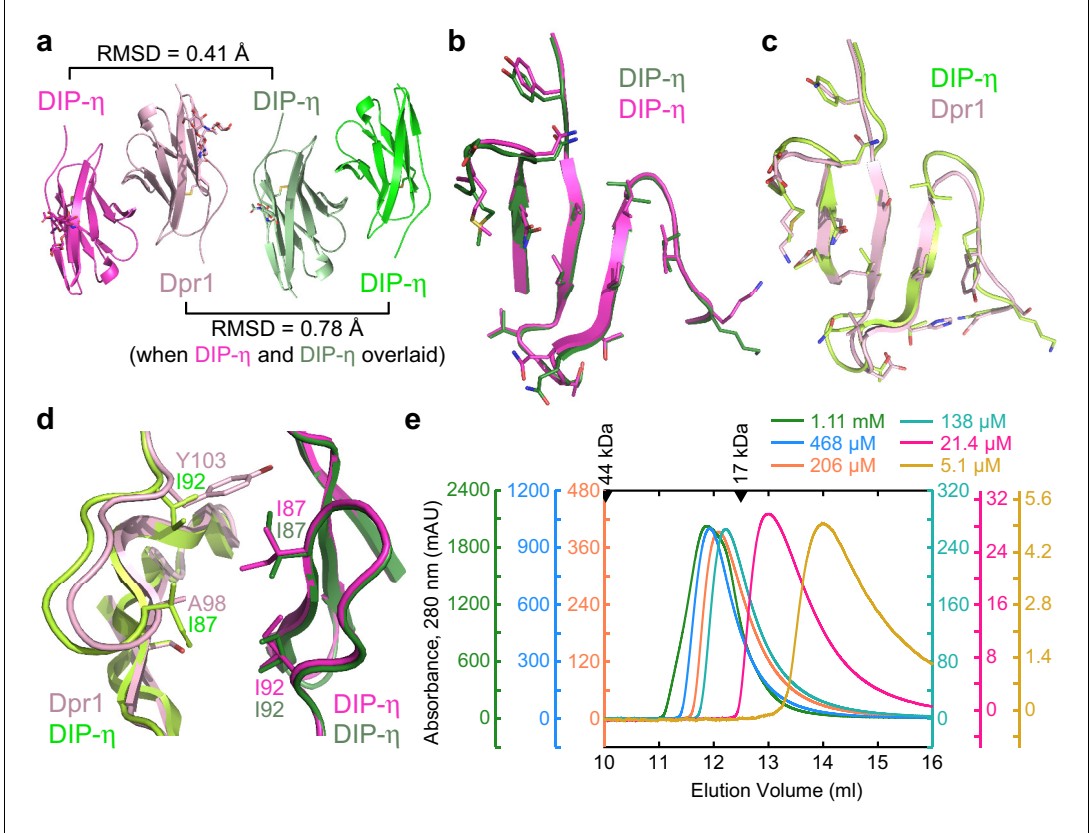

**Figure 5.** Structural comparison of DIP-η homodimer complex with the Dpr1–DIP-η heterocomplex. (a) Side-by-side aligned views of the hetero- and homophilic complexes of DIP-η. (b–c) The *GFCC'C'* faces involved in the heterophilic (magenta and pink) and homophilic (dark green and green) complexes. (d) DIP-η can accommodate binding both DIP-η and Dpr1 by rearranging the rotameric states of its interface residues. (e) Gel filtration chromatography of DIP-η IG1 at six concentrations. DIP-η is in a fast-exchange dimer-to-monomer equilibrium in the mid-micromolar range. The chromatograms are drawn at different scales shown at both sides of the plot. Path length of the UV flow cell is 0.2 cm. Elution volumes for gel filtration standards are labeled with filled triangles above the chromatograms. DIP-η peak elution positions are plotted against concentration in *Figure 5—figure supplement 1c*.

DOI: https://doi.org/10.7554/eLife.41028.013

The following figure supplement is available for figure 5:

**Figure supplement 1.** Structural comparison of Dpr1–DIP-η and DIP-η–DIP-η complexes.

DOI: https://doi.org/10.7554/eLife.41028.014

suggest that Dprs that are highly similar in sequence (≥70% identity) are unlikely to be differentiated in their DIP binding, but in time and place of expression.

The high-resolution Dpr10–DIP-α structure also allowed us to observe an ordered, near-complete N-linked glycan at the interface (*Figure 6C*). As we use lepidopteran cells to express Dprs and DIPs, the glycan structure and composition in our structure likely match the native insect Dpr/DIP glycans. The structure shows that the first N-acetyl glucosamine (NAG) is fucosylated at both the third and sixth carbon positions – commonly observed in arthropods but not in mammals. The glycan linked to Asn82 in Dpr10, which is present in seven out of 21 Dprs, is ordered as it inserts itself into a groove on the DIP-α surface, and adds a further 440 Å² area to the buried surface area (total area: 2,240 Å²). While the energetic contribution of the glycan is yet to be determined, biochemical studies of Dprs and DIPs may benefit from over-expression in eukaryotic, and specifically arthropod cell lines, due to native-like glycosylation.

During our work with Dpr10 and DIP-α, we detected DIP-α homodimerization with ECIA (*Figure 6e*, upper left corner). DIP-α homodimers are also observed via size-exclusion chromatography, similar to DIP-η, in the mid-micromolar range (*Figure 6—figure supplement 1c*). The heterophilic Dpr6–DIP-α interaction, with a $K_D$ of 0.37 µM, is stronger than homodimerization of DIP-α, as

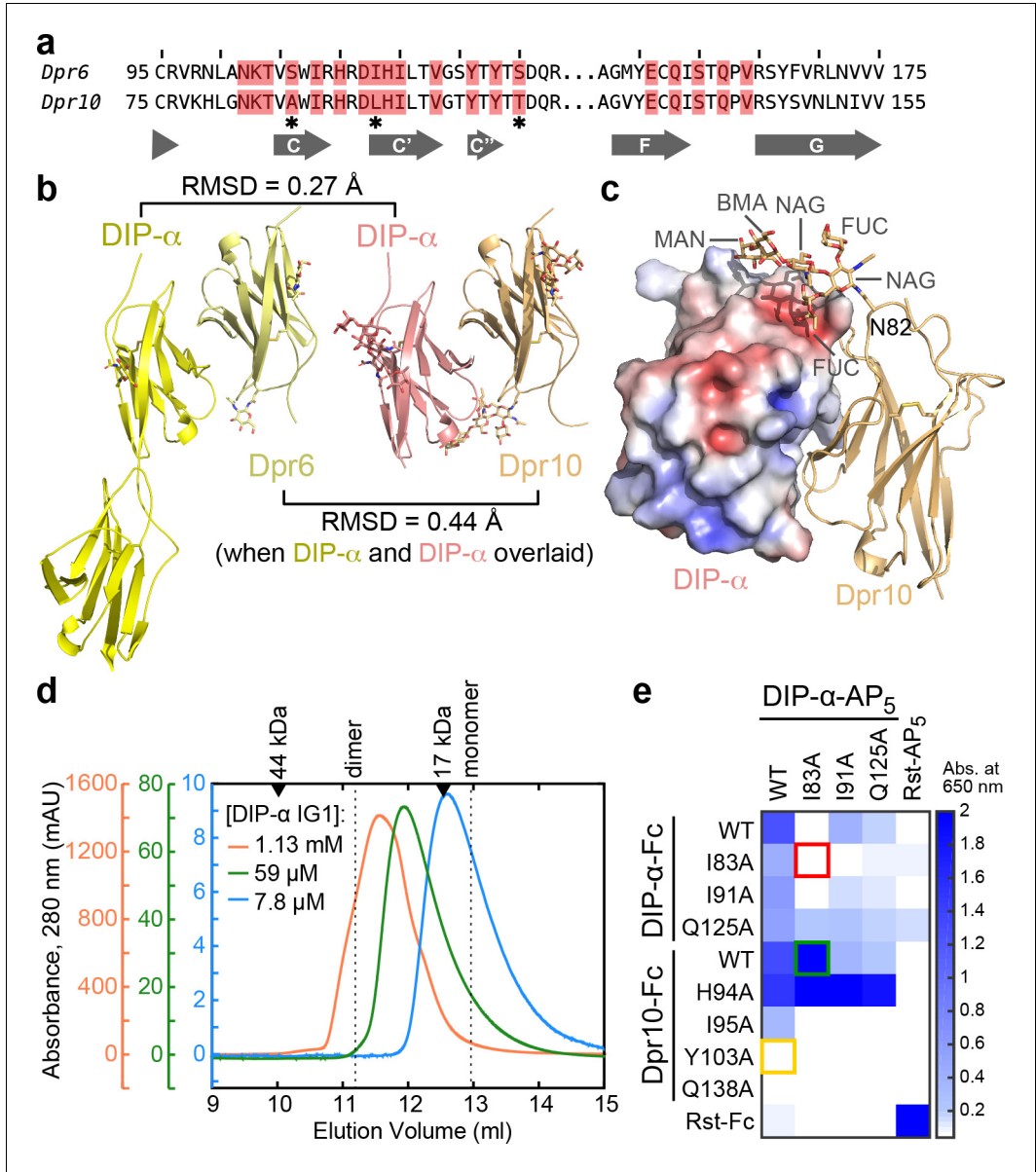

**Figure 6.** Structural description and engineering of the Dpr10–DIP-α complex. (a) Comparison of Dpr6 and Dpr10 IG1 sequences. Red boxes indicate Dpr residues within 4 Å of DIP-α in the heterocomplexes. * indicates the three variable residues between Dpr6 and Dpr10 at the interface. (b) Side-by-side view of Dpr6–DIP-α and Dpr10–DIP-α complex structures. RMSD values are reported for Cα atoms only. (c) N-linked glycan involvement at the Dpr10–DIP-α interface. DIP-α is depicted as a surface colored by electrostatic potential, and Dpr10 is in cartoon representation with the Asn82-linked glycan drawn as sticks. (d) Gel filtration chromatography of DIP-α IG1 at three DIP-α concentrations. DIP-α is in a fast-exchange dimer-to-monomer equilibrium in the mid-micromolar range. The chromatograms are drawn at different scales shown to the left of the plot. Path length of the UV flow cell is 0.2 cm. (e) ECIA screening of single-site mutants of DIP-α and Dpr10 with modified homophilic and heterophilic affinities. See main text for descriptions of the red, green and orange boxes.

DOI: https://doi.org/10.7554/eLife.41028.016

The following figure supplements are available for figure 6:

**Figure supplement 1.** Structural comparison of Dpr6–DIP-α and Dpr10–DIP-α heterodimeric complexes.
DOI: https://doi.org/10.7554/eLife.41028.017

**Figure supplement 2.** Interactions of DIP-α[I83A] and Dpr10[Y103A].
DOI: https://doi.org/10.7554/eLife.41028.018

mixing DIP-α with stoichiometric amounts of Dpr6 creates only heterophilic complexes observable on gel filtration columns, breaking up weaker DIP-α homodimers, and crystal trials including Dpr6 and DIP-α yield heterocomplex crystals (*Carrillo et al., 2015*).

As DIP-α and DIP-η can form homophilic and heterophilic interactions, genetic studies using mutations at their binding interfaces cannot unambiguously conclude whether homo- or heterophilic binding activity of these DIPs contribute to the function studied. To create molecular tools that can test the importance of this interface, and to possibly distinguish between both activities, we mutated DIP-α and its binding partner Dpr10 and searched for mutations that preferably break homophilic and/or heterophilic binding (*Figure 6e*). Using ECIA, we demonstrated that DIP-α I83A mutant (*Figure 6e*, red box) can no longer form homophilic dimers, but still has affinity towards Dpr10 (green box). Titrations using ECIA show that the I83A mutant appears to have a 7.8-fold *higher* affinity for Dpr10 than WT DIP-α (*Figure 6—figure supplement 2a*). This is in contrast to an expected loss of affinity based on our SPR data: the I83A mutation causes 700-fold loss of binding to Dpr6, which is a close paralog of Dpr10. In the context of a highly oligomerized or clustered distribution of DIP-α molecules, such as in ECIA or at the site of a cellular adhesion, the weak *cis* DIP-α homodimerization will successfully compete with the *trans* heterodimer, depressing heterophilic affinity. DIP-α I83A, which cannot homodimerize, is free to interact with Dpr10, and therefore appears to have high affinity for Dpr10.

Among the mutants tested in the mutational analysis of the Dpr10–DIP-α interface (*Figure 6e*), the Dpr10 mutant Y103A abolishes DIP-α binding (orange box), and therefore can be used to study the Dpr10–DIP-α complex function, without effecting DIP-α homodimerization.

## Dpr10 and DIP-α in the establishment of neuromuscular circuitry

The Drosophila larval neuromuscular system consists of 35 motor neurons, which innervate 30 muscles within each hemisegment, forming an invariant circuit that is ideal for delineating genetic mechanisms underlying synaptic connectivity. Although numerous screens and studies have been conducted to uncover potential connectivity molecules (*Aberle et al., 2002*; *Banovic et al., 2010*; *Liebl et al., 2003*; *Mosca et al., 2012*; *Nose, 2012*), we still lack a complete understanding of how a motor neuron chooses its appropriate muscle target(s). This critical gap in knowledge led us to investigate whether Dprs and DIPs have roles in neuromuscular development and specifically in synaptic partner choice. In a prior study, we found that several Dprs and DIPs, including Dpr11 and one of its interacting partners, DIP-γ, were expressed in motor neurons. Importantly, Dpr11 and DIP-γ are required for normal motor neuron terminal growth (*Carrillo et al., 2015*). Although the process of neuromuscular junction (NMJ) expansion does not reflect initial synaptic connectivity, the same Dpr–DIP pair is also required for connectivity in the optic lobe. Thus, we delved deeper into Dpr and DIP function at the NMJ.

In the fly larval neuromuscular system, muscles are innervated by multiple motor neurons. The majority of these motor neurons are the class one type, and these can be further subdivided into 1b (big) or 1 s (small) indicative of their terminal, or bouton, sizes. Several additional key factors differentiate 1b and 1 s motor neurons: most 1b motor neurons innervate single muscle targets whereas 1 s motor neurons innervate subgroups of muscles (dorsal, lateral and ventral muscles) and the amount of subsynaptic reticulum (SSR) surrounding 1b boutons is much greater than 1 s. Here we focus on one 1 s motor neuron, the MNISN-1s, that innervates the dorsal muscles (*Hoang and Chiba, 2001*; *Landgraf et al., 2003*). In a concurrent study, we demonstrate that *DIP-α* is expressed in MNISN-1s, and a DIP-α binding partner, Dpr10, is expressed postsynaptically in muscles. Importantly, we discovered that these interactors are absolutely required for the proper connectivity between MNISN-1s and the postsynaptic muscle target muscle 4, and partially required for connectivity to muscles 3 and 20 (*Ashley et al., 2019*). Further analysis of a *DIP-α* mutant revealed that the remaining MNISN-1s muscle connections were still present; thus, highlighting the specificity inherent in connectivity codes, even within a single neuron, and the potential requirement for combinatorial interactions between cell surface proteins for establishing synaptic partner matching. As the muscle 4 (m4) connection was the most sensitive to loss of *DIP-α* (*Ashley et al., 2019* and *Figure 7g*), we utilized this phenotype to examine if Dprs and DIPs with altered specificities could provide additional insight into our understanding of circuit wiring.

In addition to the two DIP-α binding partners, Dpr6 and Dpr10, observed in the first application of the ECIA strategy (*Özkan et al., 2013*), we have demonstrated here the DIP-α homophilic binding

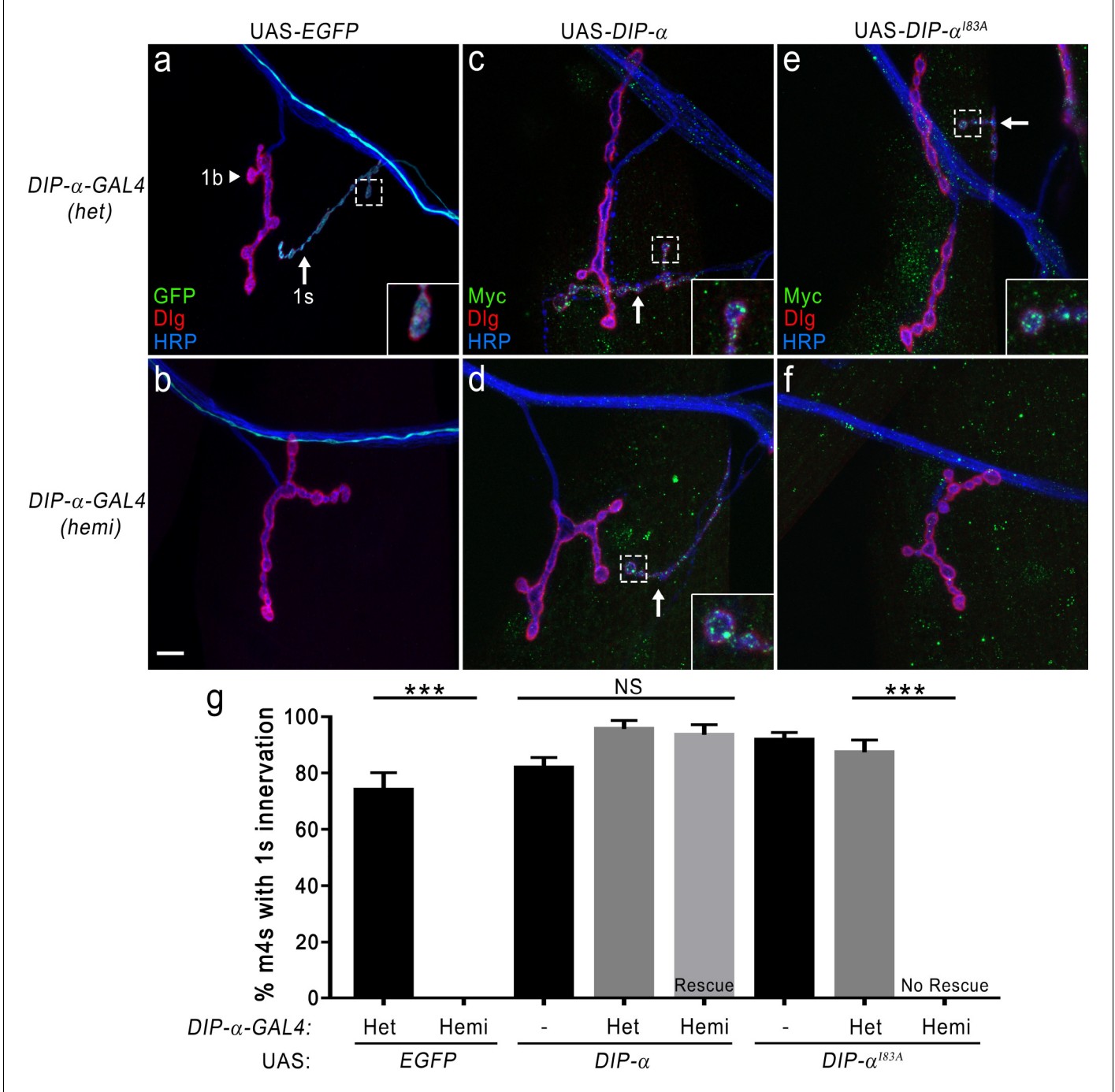

**Figure 7.** DIP-α–DIP-α interactions are required for proper MNISN-1s innervation of m4. (**a**) *DIP-α* is expressed in MNISN-1s (green) neurons. The *DIP-α-GAL4* allows for utilization of the UAS/GAL4 system and this gene trap is also a null allele (*Ashley et al., 2019*). In *DIP-α-GAL4* heterozygous (het) larvae, both 1b (arrowhead) and 1 s (arrow) terminals are present on m4. (**b**) Removal of *DIP-α* results in loss of MNISN-1s innervation of m4. The MNISN-1s axon is still visible (green) and continues to innervate other dorsal muscles. These hemizygous male larvae retain *GAL4* expression under the control of the endogenous *DIP-α* promoter. (**c**) Overexpression of UAS-*DIP-α*-Myc (shortened to UAS-*DIP-α*) does not affect innervation of m4 in a heterozygous *DIP-α-GAL4* background. DIP-α localizes to the 1 s terminals (green in inset; *Ashley et al., 2019*). Note that DIP-α protein is labeled with anti-Myc. Green signal on muscles represents non-specific labeling of anti-Myc (see *Figure 7—figure supplement 1e*). (**d**) The *DIP-α* loss-of-function phenotype is rescued by reintroducing a UAS-*DIP-α* transgene in cells that normally express DIP-α. (**e**) UAS-*DIP-α*[I83A] expression does not alter innervation of m4 and DIP-α[I83A] localizes normally within the 1 s terminals (inset). (**f**) Expression of UAS-*DIP-α*[I83A] fails to rescue the *DIP-α* loss-of-function phenotype (no 1 s innervation of m4). (**g**) Quantification of 1 s innervation of m4. Heterozygous background contains a single wild-type copy of *DIP-α*, while the hemizygous background only contains the loss-of-function allele. Expression of UAS-*DIP-α* completely rescues the loss-of-function

*Figure 7 continued on next page*

*Figure 7 continued*

phenotype, while expression of the UAS-*DIP-α*$^{I83A}$ does not. Control UAS transgene background (no GAL4) does not affect m4 innervation. n: See *Figure 7—source data 1*. ***p<0.0001. Calibration bar, 10 μm.

DOI: https://doi.org/10.7554/eLife.41028.019

The following source data and figure supplement are available for figure 7:

**Source data 1.** Source data for *Figure 7—figure supplement 1a, b and f*.

DOI: https://doi.org/10.7554/eLife.41028.021

**Source data 2.** Source data for *Figure 7g*.

DOI: https://doi.org/10.7554/eLife.41028.022

**Figure supplement 1.** Loss of DIP-α homophilic interactions does not affect expression or localization of DIP-α$^{I83A}$ and innervation of m4 is not a sex-dependent variable.

DOI: https://doi.org/10.7554/eLife.41028.020

(*Figure 6e*). This new interaction raised the important question: Is DIP-α homodimerization required for proper wiring of the neuromuscular system? In our concurrent study, we showed that removal of *DIP-α* leads to the lack of MNISN-1s innervation of m4 (*Ashley et al., 2019* and *Figure 7a,b,g*). We used a *DIP-α-GAL4* gene trap which serves the dual purpose of a loss-of-function (LOF) allele and a GAL4 driver in the bipartite GAL4/UAS system. Also, these studies take advantage of *DIP-α* being an X-linked gene since *DIP-α-GAL4* heterozygous females can be used as controls and hemizygous males represent null mutants. Utilizing this approach, we can rescue the LOF *DIP-α* phenotype by expressing a wild-type UAS-*DIP-α* in cells which normally express *DIP-α*, including MNISN-1s (*Ashley et al., 2019* and *Figure 7d,g*). We favor a model whereby transsynaptic Dpr10–DIP-α interactions mediate connectivity, which we set out to demonstrate using single-site mutations that break the interaction. For this purpose and to tease apart roles for homophilic and heterophilic interactions, we constructed a *DIP-α* mutant I83A (*DIP-α*$^{I83A}$) which abolishes homophilic binding but does, at least partly, retain the Dpr10 interaction (*Figure 6e* and *Figure 6—figure supplement 2a*). As shown in *Figure 7d and g*, expression of UAS-*DIP-α* in cells that normally express DIP-α is able to rescue the loss of *DIP-α* phenotype confirming our previous finding. However, when the same experiment is repeated with UAS-*DIP-α*$^{I83A}$, we no longer observe rescue (*Figure 7f,g*), suggesting that the DIP-α interaction interface we identified is required for the connectivity of MNISN-1s to m4, and that the connectivity might depend on homodimerization activity of DIP-α. These results are not due to changes in DIP-α$^{I83A}$ expression since the mutant and wild type DIP-α are expressed at similar levels (*Figure 7—figure supplement 1a*). Also, to confirm that there are no inherent sex differences in the formation of these terminals, we scored female and male heterozygous transgene controls and found no significant differences in m4 innervation (*Figure 7—figure supplement 1b*). Overall, this suggests a multistep process of m4 innervation requiring both a Dpr10–DIP-α interaction as well as a homomeric DIP-α–DIP-α interaction.

In our concurrent study describing the role of Dpr10–DIP-α interactions in wiring of the neuromuscular circuit, we found that overexpression of UAS-*dpr10* in muscles caused the partial loss of MNISN-1s innervation of m4, similar to the *dpr10* mutant phenotype (*Ashley et al., 2019*). We sought to determine if this gain-of-function (GOF) phenotype was a direct consequence of overexpressed Dpr10 interacting with endogenous DIP-α. As discussed above, the Dpr10 mutant Y103A (hereafter denoted *Dpr10*$^{Y103A}$) is unable to bind DIP-α (*Figure 6e*), providing an ideal tool for exploring this GOF phenotype. Unlike muscle overexpression of UAS-*Dpr10* (*Figure 8a*), similar high level expression of UAS-*dpr10*$^{Y103A}$ in muscles does not affect MNISN-1s innervation of m4 (*Figure 8b,c*), suggesting that the Dpr10–DIP-α interaction is an integral component of the Dpr10 GOF phenotype. This GOF phenotype is dependent on the levels of UAS-*dpr10* overexpression, as mild muscle expression does not reveal the GOF phenotype (*Figure 8—figure supplement 1*). To address the possibility of an unexpected downstream effect of the Dpr10$^{Y103A}$ mutant, we showed that the Y103A mutation does not affect the interaction of Dpr10 with cDIP (*Figure 6—figure supplement 2b*); however, we cannot rule out effects mediated by unknown binding partners of Dpr10. Furthermore, we reasoned that if overexpressed Dpr10 is acting through DIP-α, partial loss of *DIP-α* should exacerbate the GOF UAS-*dpr10* phenotype while overexpression of UAS-*dpr10*$^{Y103A}$ should be insensitive to DIP-α levels. Indeed, only overexpression of UAS-*dpr10* is sensitive to DIP-α levels (*Figure 8c*), supporting a role for endogenous DIP-α interaction with overexpressed Dpr10.

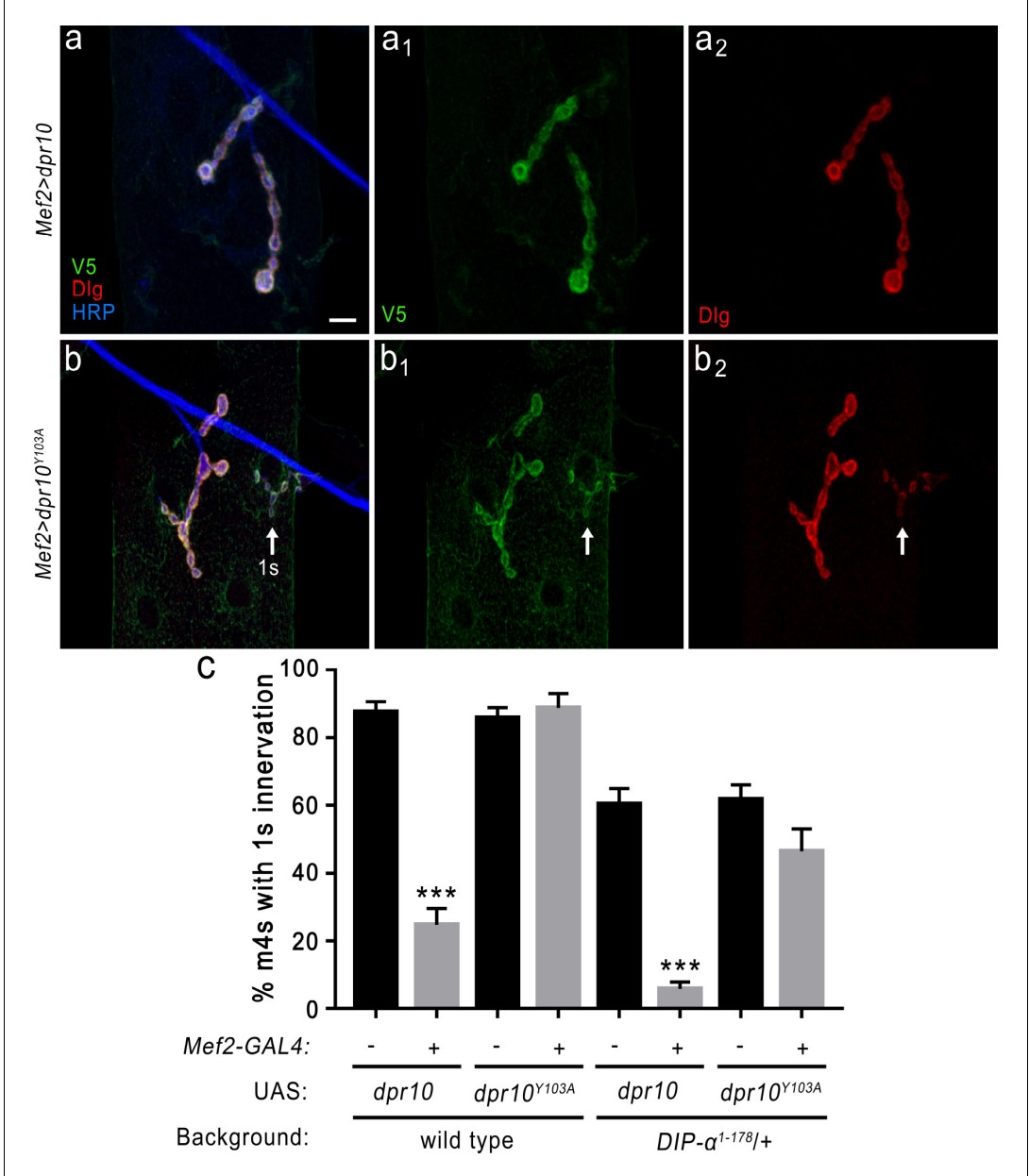

**Figure 8.** DIP-α is required for the loss of MNISN-1s innervation of m4 when overexpressing Dpr10 postsynaptically. (a) Loss of MNISN-1s innervation of m4 due to overexpression of UAS-*dpr10-V5* (referred to in the figure as UAS-*dpr10*) in muscles with the *Mef2-GAL4* driver. Dpr10 is localized specifically to the postsynaptic membrane (green) and co-localizes with Dlg, a postsynaptic membrane marker (red). Anti-HRP (blue) labels all neuronal membrane. (a$_1$) and (a$_2$) show the individual Dpr10 and Dlg channels, respectively. Note that only 1b terminals are present. Also, the Dpr10 protein is labeled with anti-V5. (b) Muscle overexpression of a Dpr10 variant (UAS-*dpr10$^{Y103A}$*) that is incapable of binding DIP-α does not affect m4 innervation. Both 1b and 1 s (arrow) terminals are present on m4. The 1b and 1 s terminals are easily distinguished by size and staining intensity of Dlg (b$_2$) (see *Materials and methods*). (c) Quantification of 1 s innervation of m4. Overexpression of wild type UAS-*dpr10* transgene results in 25% of m4s innervated by MNISN-1s compared to 89% innervation when overexpressing UAS-*dpr10$^{Y103A}$* which is unable to bind DIP-α. n: See *Figure 8—source data 1*. ***p<0.0001. Calibration bar, 10 μm.

DOI: https://doi.org/10.7554/eLife.41028.023

The following source data and figure supplement are available for figure 8:

**Source data 1.** Source data for *Figure 8c*.
DOI: https://doi.org/10.7554/eLife.41028.025
**Source data 2.** Source data for *Figure 8—figure supplement 1*.
DOI: https://doi.org/10.7554/eLife.41028.026

*Figure 8 continued on next page*

*Figure 8 continued*

**Figure supplement 1.** Weak and strong GAL4 expression of the UAS-*dpr10* transgene.
DOI: https://doi.org/10.7554/eLife.41028.024

## Discussion

Recent advances in connectomics and transcriptomics have the potential to enhance our mechanistic understanding of neuronal wiring, especially if such datasets can be matched by accurate neuronal protein interaction datasets, and a structural and evolutionary understanding of how common molecular tools across animal taxa have been repeatedly used, and regularly expanded to create more complex neuronal networks. Previous evidence shows that Dprs and DIPs may be representative of neuronal surface proteins that have expanded in the arthropod line to help wire complex but stereotyped brains.

The interaction network created by the Dprs and DIPs demonstrates how gene duplication events have led to diversity in molecular recognition and function in neuronal surface molecules. While the distant gene duplication events have given rise to the five Dpr and five DIP subclasses and have resulted in specialization of interactions, the more recent duplication events have only created mostly redundant molecular interactions. A comprehensive analysis of other arthropod Dprs and DIPs may reveal evolutionary forces that have resulted in repeated gene duplications in these families, and it is intriguing to speculate that the complexity of neural networks and the numbers of Dprs, DIPs and other neuronal surface receptors may correlate in arthropod species.

The Dpr and DIP complex structures we have determined show a two-fold pseudo-symmetric architecture. Here, we also show the presence of DIP-η and DIP-α homodimers in solution and present a symmetrical DIP-η homodimer structure that closely mimics heterodimeric Dpr–DIP complexes. This raises the question of whether the homophilic or the heterophilic interaction evolved first. Since Dpr and DIP IG1 sequences can be aligned with identities well above any random IG domain sequences, and Dpr and DIP IG1 domains are nearly identical in structure (RMSD values $\leq 1$ Å), we believe that Dpr and DIP IG1 domains may be the result of an ancient duplication event of a homodimeric IG domain. Following this logic, the DIP-η and DIP-α complexes may represent homodimers that were retained through multiple gene duplications. As heterophilic binding allows for higher diversity in neuronal recognition than homophilic would (i.e. $21 \times 9$ possible heterodimers > 30 possible homodimers), heterophilic binding must have been favored for specifying neuronal connections in complex structures such as the fly optic lobe. This is corroborated by our observations that heterodimers have higher affinities than the homodimers.

The observations we report here, including the lack of intracellular regions and the flexible nature of the ectodomain, have led us to believe that Dprs and DIPs may not be signaling receptors, and would require binding to co-receptors or secreted ligands for relaying signal to the cytoplasm upon formation of homo or heterodimers. It is also unclear if *cis* dimers can form, and signal. As *cis* dimers would inhibit productive *trans* cell-adhesive structures, their presence has significant functional relevance. We believe that interdomain flexibility and long low-complexity 'stalk' regions linking the IG domains to the membrane would enable *cis* dimerization for homodimeric DIPs, such as DIP-α and DIP-η. In fact, our SPR experiments where DIP-η is captured on solid support at high densities reports much higher apparent $K_D$ values for the Dpr1–DIP-η interaction (23 µM vs 4.0 µM measured when non-dimerizing Dpr1 is captured on SPR chip; *Figure 4—figure supplements 1,3*), as the *cis* DIP-η homodimer formation on the chip likely competes with Dpr1 binding. The *cis* homodimerization may actually be the result of a strategy to inhibit cellular adhesions resulting from relatively weak *trans* interactions, which would not be able to overcome the *cis* homodimers. This would lead to more stringent selectivity for intercellular interactions, and would prevent non-specific synapses. We examined these interactions using engineered mutations in the NMJ, and found evidence for functional relevance for both *cis* homodimeric and *trans* heterodimeric interactions, supporting our view.

The requirement of the homomeric DIP-α–DIP-α interaction for proper synaptic targeting presents a layer of complexity to what at first appearance was a straightforward binary model. We now know that DIP-α is required for proper synapse wiring, as a wild type UAS-*DIP-α* transgene in the mutant background can restore connectivity. However, when we introduce a UAS-*DIP-α*

transgene with a mutation that breaks the DIP-α–DIP-α interaction in the same mutant background, the mutant form is unable to rescue the loss of connectivity. This does not appear to be a trafficking defect, as DIP-α$^{I83A}$ appears at similar wild-type levels in 1 s terminals as it does on other muscles (*Figure 7—figure supplement 1c,d*). DIP-α$^{I83A}$ binds Dpr10, so we cannot rule promiscuous binding of DIP-α$^{I83A}$ to Dpr10 on other muscles; however, overexpression of UAS-*DIP-α$^{I83A}$* with either *DIP-α-GAL4* or *Eve-GAL4*, which also drives in MNISN-1s, does not reveal a GOF phenotype (*Figure 7—figure supplement 1f*). Instead, our data support a model in which weak *trans* interactions with other molecules are resisted by homodimeric DIP-α complexes. This mode of targeting would allow for motor neuron growth cones to bypass non-specific or very weak interactions on non-target muscles and only synapse on bona-fide muscle targets. Interestingly, our concurrent study demonstrates that Dpr10 is expressed in specific muscles during embryonic development synchronous with growth cone exploration of those muscles, and thus overcome DIP-α homodimerization in favor of the stronger Dpr10–DIP-α heterodimer.

*Note added in proof:* During the late revision stages of our manuscript, two articles from the Shapiro, Honig and Zipursky groups were published (*Cosmanescu et al., 2018*; *Xu et al., 2018*). The results in our manuscript and the accompanying manuscript (*Ashley et al., 2019*) are in general agreement. The structures presented here and in Cosmanescu et al. show a conserved mode of binding, now observed crystallographically across three DIP homodimers and five Dpr–DIP heterodimers. The conservation of the hydrophobic core and the variable polar periphery is another shared observation. The amino acids chosen to disrupt DIP-α and Dpr10 complexes, DIP-α I83 and Dpr10 Y103, were common to both studies. Finally, both sets of studies demonstrate phenotypes when DIP-α homodimers or Dpr10–DIP-α heterodimers are affected via mutagenesis.

One point of difference is in the SPR-measured affinities of heterophilic Dpr–DIP complexes. Our reported $K_D$ values for the Dpr6–DIP-α, Dpr11–DIP-γ and Dpr1–DIP-η interactions are 6, 7, and 21-fold lower (i.e. interactions are stronger), respectively, than those of Cosmanescu et al., and as a result, these heterodimer affinities are much stronger than the homodimer affinities reported by both manuscripts. We do not believe that the disparities for heterodimeric affinities are due to the presence of additional IG domains included in SPR experiments in Cosmanescu et al., since these domains do not contribute structurally and energetically to binding as we have demonstrated initially via SPR in *Carrillo et al., 2015*. Instead, we show that DIP homodimer formation may cause SPR experiments to underestimate heterodimeric affinities (i.e. over-report $K_D$ values) due to competition between the two modes of binding, which we endeavored to remedy in our study. The interactions we identified with ECIA for DIP-ζ, -η and -θ which were not detected in Cosmanescu et al. may have been affected by this artifact during SPR experiments. The measurement of accurate affinities at overlapping homo- and heterophilic binding sites remains a significant challenge, including for Dprs and DIPs.

## Materials and methods

### Phylogenetics
The regions containing immunoglobulin domains from the *D. melanogaster* Dpr and DIP sequences were aligned using MUSCLE (*Edgar, 2004*). The phylogenetic analysis was performed using PhyML (*Guindon et al., 2010*) and the phylogenetic trees were drawn with SeaView (*Gouy et al., 2010*).

### Protein expression and purification
All large-scale Dpr and DIP protein expression was done using the baculoviral expression system. Constructs were cloned into the baculoviral transfer vector pAcGP67A (BD Biosciences) and its variants, followed by co-transfection with linearized BestBac 2.0 baculoviral DNA (Expression Systems, 91–002) into Sf9 cells, using Trans-IT Insect (Mirus Bio) or Cellfectin II (Thermo Fisher, 10362–100) as the transfection reagent according to manufacturers' specifications. For protein expression, High Five cells (BTI-TN-5B1-4) grown in Insect-XPRESS (Lonza, 12-730Q) were infected at $2 \times 10^6$ cells/ml density, and conditioned media were collected at 48–66 hr post-infection for purification of secreted proteins. All proteins expressed were designed to have C-terminal hexahistidine tags for purification via immobilized metal affinity chromatography.

Proteins were purified from the media using a protocol that precipitates unknown metal chelators in media by adding 50 mM Tris pH 8.0, 5 mM $CaCl_2$ and 1 mM $NiCl_2$, followed by removal of the precipitate and batch pull-down of Dprs and DIPs via Ni-NTA Agarose beads (QIAGEN, catalog no. 30250). All proteins were further purified on size-exclusion columns (GE Healthcare), Superdex 75 10/300 for single-domain constructs and Superdex 200 Increase 10/300 for two- or three-domain constructs, and buffer exchanged into the final buffer, HEPES-buffered saline or HBS (10 mM HEPES, pH 7.2, 150 mM NaCl).

Proteins that need to be captured on streptavidin coated SPR chips were produced with C-terminal Avi- and His-tags. The Avi-tagged proteins were biotinylated after protein purification using BirA biotin ligase, followed by a second size-exclusion chromatography step.

## Size-exclusion chromatography of Fast-Exchange DIP homodimers

DIP homodimers can be observed on small-zone size-exclusion (gel filtration) chromatography (SEC) runs. Due to fast kinetics of association and dissociation, homodimerizing DIPs run as single peaks in these chromatography runs, as the timescale of the chromatography experiment (minutes to an hour) is much longer than the timescales of monomer-dimer conversions (1 second or less) as observed in SPR experiments (Stevens, 1989; Wilton et al., 2004). While there is no explicit mathematical model for small-zone SEC for fast-kinetics oligomers (Stevens, 1989), simulations can accurately predict elution profiles. Here, we make simplifications to plot binding isotherms: (1) we ignore diffusion and dispersion terms, (2) and that the elution position is given by the peak's highest point. Based on observed dimer and monomer elution velocities, measured elution volumes can be converted to dimer fraction: Since the flow rate and the column volume is constant, 'velocity' can be thought as (Elution volume)$^{-1}$. Therefore, the elution volume of a mixed monomer-dimer sample will be,

$$V_{elution,\,mixed}^{-1} = f_{dimer} \times \left(V_{elution,\,dimer}\right)^{-1} + (1 - f_{dimer}) \times \left(V_{elution,\,monomer}\right)^{-1}$$

which gives the dimer fraction, $f_{dimer}$.

Protein concentration vs. dimer fraction was fit to a binding isotherm with the following formula in MATLAB:

$$f_{dimer} = \frac{2\,[\text{Dimer}]}{2\,[\text{Dimer}] + [\text{Monomer}]} = \frac{4[\text{DIP}]_{\text{total}} + K_D - \sqrt{K_D^2 + 8K_D[\text{DIP}]_{\text{total}}}}{4[\text{DIP}]_{\text{total}}}$$

Due to simplifications and assumptions as mentioned above, and especially uncertainties in pure dimer and monomer elution volumes, we choose to provide a range, rather than a single $K_D$ value in the main text. Dissociation constant estimated from gel filtration profiles for DIP-η (23 μM) proved to be within 1.6-fold of the dissociation constant measured with SPR (14 μM) showing the utility of the method (*Figure 5—figure supplement 1*).

## Protein crystallization and structure determination

Proteins were crystallized using the sitting-drop vapor diffusion method with a Mosquito robot (TTP Labtech) at 21°C, using 100 nl protein +100 nl crystallant drops against 50 μl crystallant reservoir. Proteins used for crystallization were dissolved in HBS, unless noted otherwise.

*Dpr11–DIP-γ.* Crystals for the complex of Dpr11 IG1 with DIP-γ IG1 +2 were grown from 15 mg/ml protein sample using 0.1 M sodium citrate, pH 5.5, 2 M ammonium sulfate. Crystals were cryo-protected in 0.1 M sodium citrate, pH 5.5, 2.2 M ammonium sulfate, 30% glycerol and vitrified in liquid nitrogen. X-ray diffraction data were collected at the Advanced Photon Source, beamline 23-ID-B. Crystallographic data were reduced using *XDS* (*Kabsch, 2010*), and the structure was determined by molecular replacement with *PHASER* (*McCoy et al., 2007*) using Dpr6 and DIP-α structures (PDB ID: 5EO9). Model refinement and building were performed with *phenix.refine* (*Afonine et al., 2012*) and *Coot* (*Emsley et al., 2010*). For model validation, we used tools in the *PHENIX* (*Adams et al., 2010*) suite, specifically provided by *Molprobity* (*Chen et al., 2010*) and *Coot*.

*Dpr1–DIP-η.* Crystals for the complex of Dpr1 IG1 with DIP-η IG1 were grown from 16 mg/ml protein sample in 10 mM HEPES pH 7.2, 350 mM NaCl, using 0.2 M lithium sulfate, 0.1 M MES, pH 6, 20% (w/v) PEG 4000. Crystals were cryo-protected in 0.15 M ammonium sulfate, 0.1 M MES, pH 5.5,

25% (w/v) PEG 4000, 25% glycerol and vitrified in liquid nitrogen. X-ray diffraction data were collected at the Advanced Photon Source, beamline 23-ID-D. Crystallographic data were reduced using *HKL2000* (*Otwinowski and Minor, 1997*). Molecular replacement, model refinement, building and validation were performed as above.

*DIP-γ only.* Crystals for DIP-γ IG1 +2 were grown from a 1:1 mixture of DIP-γ and cDIP at 15 mg/ml protein sample in the crystallant 0.15 M ammonium sulfate, 0.1 M MES, pH 5.5, 25% (w/v) PEG 4000. Crystals were cryo-protected in 0.15 M ammonium sulfate, 0.1 M MES, pH 5.5, 25% (w/v) PEG 4000, 25% glycerol and vitrified in liquid nitrogen. X-ray diffraction data were collected at the Advanced Photon Source, beamline 24-ID-E. Crystallographic data were reduced using *XDS* (*Kabsch, 2010*). Molecular replacement, model refinement, building and validation were performed as above.

*DIP-η homodimer.* Crystals for DIP-η IG1 homodimers were grown from a with 20 mg/ml protein sample in the crystallant 0.1 M sodium citrate, pH 5.5, 45% (w/v) PEG 200. Crystals were cryo-protected in 0.1 M sodium citrate, pH 5.2, 50% (w/v) PEG 200 and vitrified in liquid nitrogen. X-ray diffraction data were collected at the Advanced Photon Source, beamline 24-ID-E. Crystallographic data were reduced using *XDS*. Molecular replacement, model refinement, building and validation were performed as above.

*Dpr10–DIP-α.* Crystals were grown from a 1:1 mixture of Dpr10 IG1 and DIP-α IG1 with 15 mg/ml protein sample in the crystallant 1 M lithium chloride, 0.1 M HEPES, pH 7.0, 20% (w/v) PEG 6000. Crystals were cryo-protected in 0.2 M lithium chloride, 0.1 M Tris, pH 8.0, 22% (w/v) PEG 6000, 25% glycerol and vitrified in liquid nitrogen. X-ray diffraction data were collected at the Advanced Photon Source, beamline 23-ID-B. Crystallographic data were reduced using *XDS*. Molecular replacement, model refinement, building and validation were performed as above.

Analysis of the interaction interfaces were aided by *PyMOL* (Schrödinger LLC) and *PISA* (*Krissinel and Henrick, 2007*). RMSD calculations were done in PyMOL for all Cα atoms in each IG1 domain with zero outlier rejections. Buried surface areas are reported in the text as interface areas from each polypeptide chain combined.

## Surface Plasmon Resonance

Most SPR experiments were performed, as before (*Carrillo et al., 2015*), on a Biacore T200 using Streptavidin-coated (SA) chips from GE Healthcare. The DIP-η homodimerization experiments were performed with a Biorad ProteOn XPR36 using the low-capacity Neutravidin chips. For mutants with high dissociation constants (usually >100 µM), maximum response ($R_{max}$) values could not be determined. In such cases, estimates of dissociation constants were calculated while constraining $R_{max}$ to well-determined values measured on the same channel from wild-type measurements. All buffers included the surfactant 0.05% Polysorbitan-20 to prevent non-specific binding. Fitting to binding isotherms were done in BIAEvaluation (GE Healthcare), Prism 6 (GraphPad) or MATLAB (Mathworks) using 1:1 Langmuir binding models.

Affinity measurement for homodimeric proteins by SPR are complicated by the fact that homodimers form between analyte and ligand (on the chip), between ligand and ligand (on the chip), and between analyte and analyte (in the mobile phase). To avoid ligand-ligand homodimerization on the chip, we created a SPR chip by loading it with dilute (i.e. monomeric) DIP-η and only sparsely populated a low-capacity Neutravidin chips, approximately 100 response units on the ProteOn XPR36 system. To account for homodimerization of DIP-η molecules in the mobile phase, we assumed that free, unbound DIP concentration,

$$[\text{DIP}]_{\text{free}} = \left( -K_D + \sqrt{K_D^2 + 8K_D[\text{DIP}]_{\text{total}}} \right)/4$$

and we fit the binding isotherm to

$$f_{bound} = \frac{\text{Response}}{R_{max}} = \frac{-K_D + \sqrt{K_D^2 + 8K_D[\text{DIP}]_{\text{total}}}}{3K_D + \sqrt{K_D^2 + 8K_D[\text{DIP}]_{\text{total}}}}$$

In MATLAB, yielding a $R^2$ value of 0.97 for the fit (*Figure 5—figure supplement 1d*).

## Extracellular interactome assay (ECIA)

The assay was performed according to *Özkan et al. (2013)*. One modification to the assay was the use of a higher-expressing promoter, the constitutively active Actin5C promoter, in the S2 expression plasmids, which we believe to have resulted in higher sensitivity for the assay, and the detection of the homophilic DIP-α complex. Before performing the assay, proteins, bait and prey, were western blotted and detected using an anti-penta-His antibody coupled to iFluor488 (Genscript, A01800), and protein concentrations for any wild-type +mutant set were normalized via dilution.

## Fly strains

| Fly strain | Source |
| --- | --- |
| $w^{1118}$ | Bloomington Drosophila Stock center (BDSC) |
| Mef2-GAL4 | Gift of Hugo Bellen |
| DIP-α-T2A-GAL4 | Gift of Hugo Bellen |
| DIP-α$^{1-178}$ | Gift of Lawrence Zipursky |
| UAS-DIP-α-Myc | Gift of Lawrence Zipursky |
| UAS-DIP-α$^{I83A}$-Myc | See below |
| UAS-dpr10-V5 | Gift of Lawrence Zipursky |
| UAS-dpr10$^{Y103A}$-V5 | See below |
| UAS-2XEGFP | BDSC #6874 |
| 24B-GAL4 | BDSC #1767 |
| Eve$^{RN2}$-GAL4 | BDSC #7470 |
| BG487-GAL4 | BDSC #51634 |

## *Drosophila* genetics

The *DIP-α-T2A-GAL4* gene trap line is a null allele for *DIP-α*. As *DIP-α* is an X-linked gene, hemizygous males are *DIP-α* nulls. Female *DIP-α-T2A-GAL4* flies were crossed to UAS transgenic male flies, such that all male progeny are hemizygous for *DIP-α-T2A-GAL4* and all females are heterozygous. For controls, wildtype females ($w^{1118}$) were crossed to the same UAS transgenic males. The other GAL4 lines are not located on the X chromosome so the F1 gender had no impact on experimental outcome.

## Generation of transgenic lines

Plasmid constructs were generated by PCR amplification of existing genomic DNA sequences from the UAS-*DIP-α*-Myc and UAS-*dpr10*-V5 fly lines (gifts of Lawrence Zipursky). Both were amplified using the common primers: AATAGGGAATTGGGAATTCAGATCTAAAAGGTAGGTTCAACCAC and GAGTTCTGTGTGTATAACAAATGCTG. Using site directed mutagenesis, the point mutations were introduced into DIP-α using the following primers (lowercase represents the primer mismatch): ACCAAGGCCgcTCAAGCCATCCACGAGAACGTA and AAGGCCGACACCAAGGCCgcTCAAGC-CAT. The following primers were used to introduce mutations into Dpr10: ACCAAGGCC gcTCAAGCCATCCACGAGAACGTA and AAGGCCGACACCAAGGCCgcTCAAGCCAT. The resulting products were cloned into pUASTattB using Gibson Assembly (New England Biolabs). The resulting plasmids were sent for injection (Genetivision) and inserted into attP2 (*DIP-α*$^{I83A}$-Myc) and VK20 (*dpr10*$^{Y103A}$-V5). Once established, these lines were then crossed to DIP-α-GAL4 (gift of Hugo Bellen) or Mef2-GAL4 (Bloomington Drosophila Stock Center), respectively.

## Antibodies used

| Antibody | Concentration | Source |
| --- | --- | --- |
| Goat anti-HRP-TRITC | 1:50 | Jackson Immunological Research (Jackson) #123-025-021 |

*Continued on next page*

*Continued*

| Antibody | Concentration | Source |
| --- | --- | --- |
| Goat anti-HRP-Alexa405 | 1:50 | Jackson #123-475-021 |
| Mouse anti-Dlg | 1:100 | Developmental Studies Hybridoma Bank (DSHB) #4F3 |
| Mouse anti-V5 | 1:400 | ThermoFisher #R960-25 |
| Rabbit anti-GFP | 1:1000 | ThermoFisher #A11122 |
| Rabbit anti-Dlg | 1:40,000 | Gift of Vivian Budnik (*Thomas et al., 1997*) |
| Rabbit anti-Myc | 1:200 | Cell Signaling Technology #71D10 |
| Goat anti-Mouse-Alexa488 | 1:500 | ThermoFisher #A11029 |
| Goat anti-Mouse-Alexa568 | 1:500 | ThermoFisher #A11031 |
| Goat anti-Mouse-Alexa647 | 1:500 | ThermoFisher #A21235 |
| Goat anti-Rabbit-Alexa488 | 1:500 | ThermoFisher #A11008 |
| Goat anti-Rabbit-Alexa568 | 1:500 | ThermoFisher #A11036 |

## Larval sample preparation and quantification

Wandering third instar larvae were dissected as per Menon and Zinn (*Menon et al., 2009*). Briefly, samples were dissected on Sylgard dishes (Dow) under phosphate buffered saline (PBS: 10 mM phosphate buffer, 150 mM sodium chloride) and fixed for 30 min in 4% paraformaldehyde (1:5 dilution of 20% paraformaldehyde (Electron Microscopy Sciences) in PBS). Samples were permeabilized using PBS containing 0.05% Triton-X100 (PBST), washed three times, 15 min each, with PBST and incubated with primary antibodies overnight. Samples were washed three times, 15 min each, in PBST and then incubated in secondary antibodies (see above) for two hours. Samples were finally washed in PBST and then mounted in vectashield antifade reagent (Vector Laboratories). The presence or absence of 1 s innervation was determined using a Zeiss Axioimager equipped with a 40X plan-neofluar 1.3NA objective. The differential DLG labeling, weaker in type 1 s boutons compared to 1b boutons (*Guan et al., 1996*), allowed for detection and quantification of 1b and 1 s boutons on muscle 4. Abdominal segments A2-A4 were examined for each animal, and then pooled into the final quantification. Statistical analysis was performed using a student's T-test for pairwise comparison of genotypes using Prism software (Graphpad). For multiple comparisons, statistics were performed using one-way ANOVA with Dunnett's post hoc.

## Microscopy and image analysis

Confocal microscopy was performed on a Zeiss LSM800 confocal microscope using either a 40X/1.3NA plan-neofluar objective, or a 63 × 1.4 NA plan-apo objective. Experiments performed on the same day were imaged together using identical settings.

Analysis of immunofluorescence intensity was performed using ImageJ (NIH) software. For each section of an arbor, exactly 11 confocal slices were z-projected using the sum z-projection algorithm. The HRP signal was thresholded using the Huang setting in ImageJ to outline the boutons. The mean fluorescence signal was determined using the measure function. This value was normalized to $w^{1118}$ control samples which were processed and imaged on the same day as the overexpression experiments to account for non-specific anti-Myc labeling. Finally, the normalized intensity values of DIP-$\alpha$ and DIP-$\alpha^{I83A}$ were expressed as a percentage of DIP-$\alpha$. A Student's *t*-test was run between the two data sets, and no significant difference was found.

## Acknowledgements

We thank Sonal Nagarkar-Jaiswal, Vivian Budnik, Huge Bellen and Lawrence Zipursky for reagents, Agnieszka Olechwier, Patryk Poliński and Jing Wang for technical help, and Lawrence Shapiro, Barry Honig, Filip Cosmanescu and Yeonwoo Park for discussions. We acknowledge Michael Birnbaum for help with and access to SPR equipment, and Joseph Piccirilli for access to a Mosquito crystallization robot. This work was supported in part by National Institutes of Health (NIH) Grants R01 NS097161 (to EÖ) and K01 NS102342 (to RAC), a Klingenstein-Simons Fellowship Award in the Neurosciences

(to EÖ), a Sloan Research Fellowship in Neuroscience (to EÖ), and an NIH Molecular and Cellular Biology Training Grant T32 GM007183 (to ML-R). This research used resources of the Advanced Photon Source, a US Department of Energy (DOE) Office of Science User Facility operated for the DOE Office of Science by Argonne National Laboratory under Contract No. DE-AC02-06CH11357. We thank GM/CA@APS, which has been funded in whole or in part with Federal funds from the National Cancer Institute (ACB-12002) and the National Institute of General Medical Sciences (NIGMS) (AGM-12006). We also thank NE-CAT at APS, which has been funded by NIGMS grant P30 GM124165, and an NIH-ORIP HEI grant (S10OD021527). The study also used the Stanford Synchrotron Radiation Lightsource, SLAC National Accelerator Laboratory, which is supported by the US Department of Energy, Office of Science, Office of Basic Energy Sciences under Contract No. DE-AC02-76SF00515. The SSRL Structural Molecular Biology Program is supported by the DOE Office of Biological and Environmental Research, and by the National Institutes of Health, National Institute of General Medical Sciences (including P41 GM103393).

## Additional information

### Funding

| Funder | Grant reference number | Author |
| --- | --- | --- |
| National Institute of Neurological Disorders and Stroke | R01 NS097161 | Engin Özkan |
| National Institute of Neurological Disorders and Stroke | K01 NS102342 | Robert A Carrillo |
| Esther A. and Joseph Klingenstein Fund | | Engin Özkan |
| Alfred P. Sloan Foundation | | Engin Özkan |
| National Institute of General Medical Sciences | T32 GM007183 | Meike Lobb-Rabe |

The funders had no role in study design, data collection and interpretation, or the decision to submit the work for publication.

### Author contributions

Shouqiang Cheng, Conceptualization, Formal analysis, Validation, Investigation, Visualization, Methodology, Writing—original draft, Writing—review and editing; James Ashley, Formal analysis, Investigation, Visualization, Methodology, Writing—review and editing; Justyna D Kurleto, Meike Lobb-Rabe, Yeonhee Jenny Park, Investigation, Methodology, Writing—review and editing; Robert A Carrillo, Conceptualization, Formal analysis, Supervision, Funding acquisition, Investigation, Visualization, Methodology, Writing—original draft, Writing—review and editing; Engin Özkan, Conceptualization, Formal analysis, Supervision, Funding acquisition, Investigation, Visualization, Methodology, Writing—original draft, Project administration, Writing—review and editing

### Author ORCIDs

Robert A Carrillo http://orcid.org/0000-0002-2067-9861
Engin Özkan http://orcid.org/0000-0002-0263-6729

### Decision letter and Author response

Decision letter https://doi.org/10.7554/eLife.41028.039
Author response https://doi.org/10.7554/eLife.41028.040

## Additional files

### Supplementary files

• Transparent reporting form
DOI: https://doi.org/10.7554/eLife.41028.029

## Data availability

Structural models and diffraction data have been deposited in PDB (accession numbers: 6NRQ, 6NRR, 6NRX, 6NRW, and 6NS1).

The following datasets were generated:

| Author(s) | Year | Dataset title | Dataset URL | Database and Identifier |
|---|---|---|---|---|
| Shouqiang Cheng, James Ashley, Justyna D Kurleto, Meike Lobb-Rabe, Yeonhee Jenny Park, Robert A Carrillo | 2019 | Crystal structure of Dpr10 IG1 bound to DIP-alpha IG1 | http://www.rcsb.org/structure/6NRQ | Protein Data Bank, 6NRQ |
| Shouqiang Cheng | 2019 | Crystal structure of Dpr11 IG1 bound to DIP-gamma | http://www.rcsb.org/structure/6NRR | Protein Data Bank, 6NRR |
| Shouqiang Cheng, James Ashley, Justyna D Kurleto, Meike Lobb-Rabe, Yeonhee Jenny Park, Robert A Carrillo, Engin Özkan | 2019 | Crystal structure of DIP-eta IG1 homodimer | http://www.rcsb.org/structure/6NRX | Protein Data Bank, 6NRX |
| Shouqiang Cheng, James Ashley, Justyna D Kurleto, Meike Lobb-Rabe, Yeonhee Jenny Park, Robert A Carrillo, Engin Özkan | 2019 | Crystal structure of Dpr1 IG1 bound to DIP-eta IG1 | http://www.rcsb.org/structure/6NRW | Protein Data Bank, 6NRW |
| Shouqiang Cheng, James Ashley, Justyna D Kurleto, Meike Lobb-Rabe, Yeonhee Jenny Park, Robert A Carrillo, Engin Özkan | 2019 | Crystal structure of DIP-gamma IG1 +IG2 | http://www.rcsb.org/structure/6NS1 | Protein Data Bank, 6NS1 |

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
