## [Decision Letter]

Thank you for submitting your article "Molecular Basis of Synaptic Specificity by Immunoglobulin Superfamily Receptors in *Drosophila*" for consideration by *eLife*. Your article has been reviewed by Eve Marder as the Senior Editor, a Reviewing Editor, and three reviewers. The reviewers have opted to remain anonymous.

The reviewers have discussed the reviews with one another and the Reviewing Editor has drafted this decision to help you prepare a revised submission.

Summary:

This manuscript presents a structural study of proteins from the immunoglobulin superfamily, Dprs and DIPs, which have been reported to regulate synaptic connectivity in the *Drosophila* central and peripheral nervous systems. The authors use evolutionary, biophysical and biochemical approaches to analyze sequence similarities and differences in specific regions of Dpr/DIP heterophilic complex, including the hydrophobic core, in order to evaluate binding specificity after mutating amino acids at the interface. The authors also compare the structure of a homodimer of DIP with heterodimers between DIP and Dpr. They generate multiple variants that selectively abolish homodimerization of DIP-α, or the formation of heteromeric complexes of DIP-α with Dpr10. Using the *Drosophila* larval neuromuscular junction as a model, the authors conduct in vivo studies to further test the roles of DIP-α/Dpr10 complexes and DIP-α homodimers in neuromuscular circuit wiring. The authors highlight the function of cis-dimers of DIPs in regulating synaptic wiring, on top of the reported heterodimeric trans interactions between DIPs and Dprs.

The structural studies are extensive and outstanding and will provide useful information for the research community focusing on proteins containing immunoglobulin domains. While the quality of the structures data and analyses is generally high, the in vivo study of DIP-α/Dpr10 should be improved to better support the conclusions as outlined below.

Essential revisions:

Overall

It would greatly strengthen the work to more clearly highlight what the novel aspects are. What do we now know that we did not know before about how the DPR:DIP system works? Several strategic sentences placed throughout the text would be very helpful for the reader.

In vivo experiments

Some alternative explanations for the Dpr10 / DIP-α mutants need to be considered: what evidence exists that the only defect in Y103A is its interaction with DIP-α? The data suggest the authors' preferred model IF there are no other defects in the function / ability to interact with other downstream molecules of this particular Dpr10 mutant. Is this true? If this evidence does not exist, the authors should at least highlight that possibility more clearly. A similar argument for DIP-α I83A- the mutation removes some of the interaction ability with Dpr10. How can the authors ensure that this removal doesn't also affect the system? What if the phenotype is sensitive to Dpr10 interaction levels? Though these may be the most parsimonious explanations, they have not been conclusively shown by the data and more information should be offered or an appropriate scaling back of the conclusions and recognition of alternate explanations should be made.

The NMJ studies lack certain controls and experiment information – especially in Figure 7 and Figure 8, the data lack a wild-type control, a base loss-of-function to evaluate the phenotype, and sample sizes for all experiments. For instance, a Gal4 control or the control with UAS single transgene background should be added. What is Is innervation of muscle 4 normally? Does the heterozygous condition cause a partial phenotype? Or is that normal? Further, the authors cite that Dlg staining is used to tell the difference between Ib and Is boutons, but Figure 8A has no Dlg staining. This should be included. Though Ashley et al., is cited, it is not vetted by peer review. Therefore, additional data are needed here to assess the phenotype.

The interpretation of the results in Figure 7 is confusing. The evidence about the requirement of DIP-α's for proper targeting of MNISN-1s boutons appears later in Figure 8A,B. This reverse order makes the interpretation hard to follow in Figure 7. The authors should re-arrange the figures.

The authors do not show the effects of gain-of-function of Dpr10 or its variants in the muscle in a DIP-α hemizygous mutant background. This is the direct evidence to suggest that DIP-α is required for the loss of MNISN-1s boutons when overexpressing Dpr10. More appropriately, the authors should mutate endogenous Dpr10 to a form that does not bind to DIP-α, and test if MNISN-1s boutons are lost. The negative results after gain-of-function of a Dpr10 variant only suggest that an excessive amount of this Dpr10 variant is insufficient to affect 1s bouton targeting.

In Figure 7, a single muscle driver *Mef2*-Gal4 was used to overexpress the Dpr10-V5 and its variant, and the quantification in Figure 7C suggests that it leads to an increase of 1s bouton innervating m4. Using independent lines with stronger or weaker expression than *Mef2*-Gal may result in different outcomes and hence, might provide more information about the effects of overexpression of Dpr10 variants.

Could 1s bouton sizes and numbers be quantified to compare with wild type in Figure 7 and Figure 8? These may provide additional information about the functional influence of DIP/Dpr interactions.

A concern in Figure 8E and 8F is that UAS-DIP-α I83A may not function at all in vivo, as only negative results are reported in the paper using this transgene. Testing the effect on 1s boutons after overexpressing DIP-α I83A in a wild type background would be a way to evaluate the functional role of this variant. The expression level of DIP-α after overexpression of DIP-α I83A could also be tested for a comparison with the wild type.

Figure 7 and Figure 8. It would be helpful to indicate in the legends how many m4s were analyzed and quantified (Figure 7C and Figure 8G).

Overall, the NMJ section is written such that particular aficionados will understand it, but few others will. The authors should present a more general assessment, offering more to discuss the difference between type Ib and Is boutons (this knowledge is mostly taken for granted in the current version), and describe the DIP / Dpr reagents used (key amongst being the knowledge of what DIP-α-GAL4 is and that it is also an allele – this was unclear unless you have intimate familiarity with exactly what reagents the authors are employing). This will aid in accessibility, understanding, and proper assessment by the field of the story.

Presumably if they're looking at heterozygous versus hemizygous, this gene is on X, so the two larvae being compared are male and female. Is this appropriate? Are there any inherent sex differences in any of these terminals that could confound this interpretation?

Please include more detail in *Drosophila* genetics and stocks used in the Materials and methods section (male vs. female, het versus hemi, what lines are used, etc.). The current version is a little scant.

Structures and analyses

Table 1 and Table 2 should include statistics from the Ramachandran plot or a Molprobity score, and estimated error estimates of the structures.

Subsection “Shared and divergent features in the structures of Dpr–DIP heterocomplexes”. "Hence, different Dpr-DIP complexes can be established not only through shape complementarity between Dpr and DIP surfaces, but also by small but significant movements of the Dpr and DIP monomers with respect to each other, a mechanism not commonly recognized for related interaction pairs." Are there really small but significant movements of the Dpr and DIP monomers with respect to each other? To conclude this the authors must more accurately describe the differences between the structures.

Are the DIP Ig1s subunits truly displaced versus the DPR Ig1s or are the protein domain folds slightly different? If one superimposes the DIP Ig1 domains, is the fit of the DIPs very good and do the Ds then seem offset?

Please include how many Cα atoms were used in the structure for each superposition. Superpositions can vary depending on the number of Cα positions you use (or other atoms) and which these are.

Are there multiple copies of the complexes in the asymmetric unit of the different crystal forms? If so, how well do these superimpose?

Subsection “Molecular details of Dpr–DIP complex interfaces driving specificity”. "Yet, we also observed differences at these conserved positions at the structural level via rotameric changes and by rigid-body movements of DIPs with respect to Dprs (Figure 3C),..". Please show electron density in Figure 3C to support this statement about rotameric changes.

Results section. It is quite difficult to follow the structure-based engineering strategy. Please add a (supplementary) figure that shows the interface (red highlighted residues of interface in Figure 3B) mapped onto the ribbon diagram of a prototype Dpr:DIP structure using the view shown in Figure 3A. Please then indicate the side chains of all the residues that were mutated, e.g., residues H94 and Q126 as well as all the residues mentioned in Figure 4A-F. Please also indicate for example Dpr11 A165Y F167Y K207V, Dpr10Y103A and DIP-α I83A so that the reader can easily orientate themselves and see how all these residues localize with respect to the interface (i.e., the red highlighted residues in Figure 3B) by looking at a single figure. This is important because for instance in Figure 3D it is very hard to see His94 (Dpr1) to Glu126 (DIP-ε) and where it is located with respect to the interface.

Subsection “Energetics of the Dpr-DIP complex interface”. "the hydrophobic conserved core of the interface provides much of the energy of binding, and the periphery is likely responsible for specificity.". To evaluate this statement, the reader needs a figure as requested in the previous point.

Other points

Figure 5E is surprising. Please show the calibration markers for the SEC column. Please indicate where the dimer and the monomer are expected. What are the concentrations shown (please add to legend). Please explain why the magenta and mustard concentrations run so differently? Are these both monomers?

It is not clear in Figure 5 —figure supplement 1 Figure how the curves in Figure 5E could be used to calculate the ratio of monomer to dimer, leading to an estimated KD of 10-40 µM. This is a key point used to argue that DIP homodimers form with somewhat less affinity than Dpr:DIP heteromers.

The legend of Figure 5 —figure supplement 1 Figure states: "Dimer fraction was calculated by identifying monomeric and dimeric elution velocities and calculating monomer and dimer fractions for any given peak position." However, the trailing shoulders of the peaks in Figure 5E cannot be assumed to be dimer or monomer based on the elution position in the chromatogram alone because of protein loading issues. Large amounts of protein will have larger trailing peaks. So, the contents of the peaks must be validated by PAGE (native gels) to demonstrate dimer, monomer or a mixture.

Likewise, Figure 6D is surprising as well. If you have a monomer-dimer equilibrium you would expect two resolved peaks (e.g., the red curve for the dimer and the blue curve for the monomer). But a mixture should show two peaks/adjacent bumps (one for the dimer and one for the monomer) not something in the middle (green curve).

Is there perhaps a typo for the green curve in Figure 6D which reads 59 μM? Please show the calibration markers for the SEC column in Figure 6D.

For all SPR data please add the model used (presumably 1:1 interaction model) to each curve so the reader can see how well the data fits the model and thus how accurate the KD are that are calculated based on the data. Please state in the legend which interaction model was used.

---

## [Author Response]

Summary:This manuscript presents a structural study of proteins from the immunoglobulin superfamily, Dprs and DIPs, which have been reported to regulate synaptic connectivity in the Drosophila central and peripheral nervous systems. The authors use evolutionary, biophysical and biochemical approaches to analyze sequence similarities and differences in specific regions of Dpr/DIP heterophilic complex, including the hydrophobic core, in order to evaluate binding specificity after mutating amino acids at the interface. The authors also compare the structure of a homodimer of DIP with heterodimers between DIP and Dpr. They generate multiple variants that selectively abolish homodimerization of DIP-α, or the formation of heteromeric complexes of DIP-α with Dpr10. Using the Drosophila larval neuromuscular junction as a model, the authors conduct in vivo studies to further test the roles of DIP-α/Dpr10 complexes and DIP-α homodimers in neuromuscular circuit wiring. The authors highlight the function of cis-dimers of DIPs in regulating synaptic wiring, on top of the reported heterodimeric trans interactions between DIPs and Dprs.The structural studies are extensive and outstanding and will provide useful information for the research community focusing on proteins containing immunoglobulin domains. While the quality of the structures data and analyses is generally high, the in vivo study of DIP-α/Dpr10 should be improved to better support the conclusions as outlined below.Essential revisions:OverallIt would greatly strengthen the work to more clearly highlight what the novel aspects are. What do we now know that we did not know before about how the D:DIP system works? Several strategic sentences placed throughout the text would be very helpful for the reader.

We thank the reviewers for these comments. We feel we have addressed their concerns to the best of our ability.

In vivo experimentsSome alternative explanations for the Dpr10 / DIP-α mutants need to be considered: what evidence exists that the only defect in Y103A is its interaction with DIP-α? The data suggest the authors' preferred model IF there are no other defects in the function / ability to interact with other downstream molecules of this particular Dpr10 mutant. Is this true? If this evidence does not exist, the authors should at least highlight that possibility more clearly.

Dpr10 was first reported in the literature as part of our large-scale protein interactome dataset, which identified DIP-α and cDIP as the only binding partners (Özkan et al., 2013). To address the possibility that Dpr10 Y103A mutation might break an interaction with cDIP, we repeated the binding assay with cDIP. Our results (the new Figure 6—figure supplement 2B) now show that Y103A does not affect cDIP binding. However, we agree with the reviewers that we cannot rule out downstream effects of this single-site mutation mediated by unknown binding partners. The text reflects this now.

A similar argument for DIP-α I83A- the mutation removes some of the interaction ability with Dpr10. How can the authors ensure that this removal doesn't also affect the system? What if the phenotype is sensitive to Dpr10 interaction levels? Though these may be the most parsimonious explanations, they have not been conclusively shown by the data and more information should be offered or an appropriate scaling back of the conclusions and recognition of alternate explanations should be made.

We agree with the reviewers. The known heterophilic binding partners of DIP-α are Dpr6 and Dpr10 – cDIP does not bind DIP-α (Özkan et al., 2013). We already show in Figure 4—figure supplement 3 that the DIP-α I83A mutation does decrease affinity towards the closely related Dpr6 binding by ~700 fold, and we expect a similar effect for Dpr10–DIP-α Ι83Α binding. The expected partial loss of affinity in the Dpr10–DIP-α I83A interaction due to the I83A mutation might illicit a (partial) phenotype.

For our revision, we conducted a titration series of DIP-α wild-type and the I83A mutant against Dpr10 using ECIA. In the context of our high-avidity binding assay designed to mimic adhesion molecule interactions between molecular clusters on two-dimensional surfaces, the DIP-α I83A mutant binds Dpr10 not weaker, but ~8x stronger than DIP-α WT. This apparent higher heterophilic affinity for I83A is likely due to abolishment of the competing DIP-α–DIP-α interaction, which is only present for DIP-α WT in the context of the highly clustered ECIA. Even though we expect DIP-α I83A to bind Dpr10 weaker, lack of competing DIP-α homodimerization allows for DIP-α I83A to retain effective binding to Dpr10 better than the WT. Unfortunately, it is very hard to predict what would happen on a neuron, and whether the SPR setup (Figure 4) or the ECIA (Figure 6—figure supplement 2) is more representative of the physiological interaction. Any DIP mutant that only removes homodimerization may effectively increase Dpr affinity depending on the avidity (clustering and local concentration) of the binding partners in the system being studied.

Based on these results, DIP-α I83A may be a valuable tool to test the functional outcomes for lack of homophilic DIP-α binding, while the heterodimer (DIP-α–Dpr10) can still form. However, we believe the safer course would be to explicitly state alternate explanations in our manuscript, as the reviewers suggest. We have changed the results and Discussion sections to reflect this.

The NMJ studies lack certain controls and experiment information – especially in Figure 7 and Figure 8, the data lack a wild-type control, a base loss-of-function to evaluate the phenotype, and sample sizes for all experiments. For instance, a Gal4 control or the control with UAS single transgene background should be added. What is Is innervation of muscle 4 normally? Does the heterozygous condition cause a partial phenotype? Or is that normal? Further, the authors cite that Dlg staining is used to tell the difference between Ib and Is boutons, but Figure 8A has no Dlg staining. This should be included. Though Ashley et al., is cited, it is not vetted by peer review. Therefore, additional data are needed here to assess the phenotype.

These controls have now been added to the manuscript. See Figure 7G and Figure 8C for updated figures with heterozygous UAS transgene controls. These controls all demonstrate the ~80% innervation frequency of muscle 4. The heterozygous DIP-α-GAL4 condition is not statistically different from wild type (p>0.5). Anti-Dlg labeling has been added to Figure 7 (formerly Figure 8).

The interpretation of the results in Figure 7 is confusing. The evidence about the requirement of DIP-α's for proper targeting of MNISN-1s boutons appears later in Figure 8A,B. This reverse order makes the interpretation hard to follow in Figure 7. The authors should re-arrange the figures.

The figures have now been rearranged and we feel this greatly improves the flow of the manuscript. We thank the reviewer for this keen observation.

The authors do not show the effects of gain-of-function of Dpr10 or its variants in the muscle in a DIP-α hemizygous mutant background. This is the direct evidence to suggest that DIP-α is required for the loss of MNISN-1s boutons when overexpressing Dpr10. More appropriately, the authors should mutate endogenous Dpr10 to a form that does not bind to DIP-α, and test if MNISN-1s boutons are lost. The negative results after gain-of-function of a Dpr10 variant only suggest that an excessive amount of this Dpr10 variant is insufficient to affect 1s bouton targeting.

The reviewer is correct; we did not show this genetic interaction. We have now included this data in Figure 8C and feel it has improved our stance that this is a Dpr10-DIP-α specific gain of function phenotype. Although we agree that mutation of endogenous Dpr10 would be ideal, there was not adequate time to perform this experiment.

In Figure 7, a single muscle driver Mef2-Gal4 was used to overexpress the Dpr10-V5 and its variant, and the quantification in Figure 7C suggests that it leads to an increase of 1s bouton innervating m4. Using independent lines with stronger or weaker expression than Mef2-Gal may result in different outcomes and hence, might provide more information about the effects of overexpression of Dpr10 variants.

We apologize for any confusion on our part. We now show that Mef2>dpr10 induces a gain-of-function phenotype that reduces the frequency of m4 innervation from 80% to 25% (Figure 8C). This phenotype is replicated with another strong muscle GAL4 line, 24B-GAL4, but not with a weak muscle GAL4 line, BG487-GAL4 (Figure 8—figure supplement 1).

Could 1s bouton sizes and numbers be quantified to compare with wild type in Figure 7 and Figure 8? These may provide additional information about the functional influence of DIP/Dpr interactions.

We thank the reviewers for making this suggestion, however, this data does not relate to synaptic connectivity and instead relates to synaptic outgrowth, which is beyond the scope of this manuscript.

A concern in Figure 8E and 8F is that UAS-DIP-α I83A may not function at all in vivo, as only negative results are reported in the paper using this transgene. Testing the effect on 1s boutons after overexpressing DIP-α I83A in a wild type background would be a way to evaluate the functional role of this variant. The expression level of DIP-α after overexpression of DIP-α I83A could also be tested for a comparison with the wild type.

The reviewer makes a good point about the DIP-α^I83A^ function, as we only show expression in a mutant background. Without including the controls, the phenotypes were unclear. Now with the appropriate controls, as suggested by the reviewer, we can show that either UAS-*DIP-α* or UAS-*DIP-α^I83A^* expression in a heterozygous mutant background (Figure 7G) or in a wild type background (Figure 7—figure supplement 1F) show no change in innervation of m4. We also compare expression levels of the two transgenes and find no significant change between them (Figure 7—figure supplement 1A). These additional experiments suggest that both constructs are expressed and trafficked normally, and that expression of UAS-*DIP-α^I83A^* does not alter innervation.

Figure 7 and Figure 8. It would be helpful to indicate in the legends how many m4s were analyzed and quantified (Figure 7C and Figure 8G).

We apologize for this oversight on our part. We have now included four tables to accompany each figure with the mean, SD and SEM for each graph, as well as number of animals and number of hemisegments quantified.

Overall, the NMJ section is written such that particular aficionados will understand it, but few others will. The authors should present a more general assessment, offering more to discuss the difference between type Ib and Is boutons (this knowledge is mostly taken for granted in the current version), and describe the DIP / Dpr reagents used (key amongst being the knowledge of what DIP-α-GAL4 is and that it is also an allele – this was unclear unless you have intimate familiarity with exactly what reagents the authors are employing). This will aid in accessibility, understanding, and proper assessment by the field of the story.

Again, we apologize for the oversight. We have now gone back and carefully explained phenotypes, controls and methods in a thorough manner.

Presumably if they're looking at heterozygous versus hemizygous, this gene is on X, so the two larvae being compared are male and female. Is this appropriate? Are there any inherent sex differences in any of these terminals that could confound this interpretation?

As stated above, the lack of clarity on our part was an oversight. We have now improved our description of this genotype as well as included male and female controls to show there is no sex specific differences in m4 innervation (Figure 7—figure supplement 1B).

Please include more detail in Drosophila genetics and stocks used in the Materials and methods section (male vs. female, het versus hemi, what lines are used, etc.). The current version is a little scant.

We did not include a final version of our Materials and methods section in our initial submission and neglected to include this with the version that went out to reviewers. We have now included a complete Materials and methods section in this version of the manuscript.

Structures and analysesTable 1 and Table 2 should include statistics from the Ramachandran plot or a Molprobity score, and estimated error estimates of the structures.

We had used a template from a prominent Structural Biology journal for our crystallographic tables. As the reviewers noticed, the tables lacked important validation criteria. This has been corrected in both tables. We thank the reviewers for pointing this out.

Subsection “Shared and divergent features in the structures of Dpr–DIP heterocomplexes”. "Hence, different Dpr-DIP complexes can be established not only through shape complementarity between Dpr and DIP surfaces, but also by small but significant movements of the Dpr and DIP monomers with respect to each other, a mechanism not commonly recognized for related interaction pairs." Are there really small but significant movements of the Dpr and DIP monomers with respect to each other? To conclude this the authors must more accurately describe the differences between the structures.

The original manuscript mentioned: “When the Dpr subunits are forced to align, the three DIP subunits are slightly misplaced, with DIP-γ (dark gray in Figure 2 and Figure 3) more distant from the other DIPs (~1.2 Å at the interface and up to 3 Å at the back face of the IG domain, Figure 3A)”. However, the most remarkable point here is that *CC’C”FG* sheets are separated from each other at different distances when the three Dpr-DIP complexes are compared. The new Figure 3—figure supplement 1B top image shows this clearly. We thank the reviewers for highlighting the need to better demonstrate this point.

Are the DIP Ig1s subunits truly displaced versus the DPR Ig1s or are the protein domain folds slightly different? If one superimposes the DIP Ig1 domains, is the fit of the DIPs very good and do the DPRs then seem offset?

The domain folds among the Dprs and the DIPs show strong matches between the different complexes. For example, the Cα atoms have rmsd values ranging from 0.59 to 0.79 Å among all DIP IG1 domains (for all Cα atoms, no outliers are rejected). This is not surprising, since the all DIPs are all very similar in sequence: DIP-α, -γ and -η do not even contain a single insertion/deletion in the multiple sequence alignment of IG1s. The complexes have much higher RMSDs with respect to each other: especially Dpr11-DIP-γ complex is 1.7-2.0 Å different from the others.

Aligning of DIP interface amino acids causes similarly high displacement between Dpr chains.

Please include how many Cα atoms were used in the structure for each superposition. Superpositions can vary depending on the number of Cα positions you use (or other atoms) and which these are.

We thank the reviewers for pointing this out. When calculating RMSDs, we use only Cα atoms, and turn off outlier rejection cycles. No outlier rejection means that all single subunit comparisons are for 100 or 101 Cα atoms, except for Dpr11 (94 resolved amino acids in crystal structure).

Are there multiple copies of the complexes in the asymmetric unit of the different crystal forms? If so, how well do these superimpose?

Of the complex structures we report, Dpr1-DIP-η complex has two complexes in the asymmetric unit (asu). The RMSD between the complexes is 0.27 Å over 203 Cα atoms. The Dpr10-DIP-α crystals also have two complexes per asu, and their RMSD is 0.36 Å over 204 Cα atoms. These low RMSD values, similar to ML coordinate error estimates, show very close matches between the NCS-related copies.

Subsection “Molecular details of Dpr–DIP complex interfaces driving specificity”. "Yet, we also observed differences at these conserved positions at the structural level via rotameric changes and by rigid-body movements of DIPs with respect to Dprs (Figure 3C),..". Please show electron density in Figure 3C to support this statement about rotameric changes.

We have added panels to Figure 3—figure supplement 1C,D in the revised manuscript to demonstrate electron density quality at these side chain positions.

Results section. It is quite difficult to follow the structure-based engineering strategy. Please add a (supplementary) figure that shows the interface (red highlighted residues of interface in Figure 3B) mapped onto the ribbon diagram of a prototype Dpr:DIP structure using the view shown in Figure 3A. Please then indicate the side chains of all the residues that were mutated, e.g., residues H94 and Q126 as well as all the residues mentioned in Figure 4A-F. Please also indicate for example Dpr11 A165Y F167Y K207V, Dpr10Y103A and DIP-α I83A so that the reader can easily orientate themselves and see how all these residues localize with respect to the interface (i.e., the red highlighted residues in Figure 3B) by looking at a single figure. This is important because for instance in Figure 3D it is very hard to see His94 (Dpr1) to Glu126 (DIP-ε) and where it is located with respect to the interface.

We share the frustration of the reviewers. The interface is a curving three-dimensional space, and a single two-dimensional image fails to show all residues of importance. Following the reviewer’s advice, we have added Figure 3—figure supplement 1 and Figure 4—figure supplement 4 to aid visualization of the amino acids at the interface.

Subsection “Energetics of the Dpr-DIP complex interface”. "the hydrophobic conserved core of the interface provides much of the energy of binding, and the periphery is likely responsible for specificity.". To evaluate this statement, the reader needs a figure as requested in the previous point.

We have improved added a figure to make this point clearer. Please see Figure 3—figure supplement 1A.

Other pointsFigure 5E is surprising. Please show the calibration markers for the SEC column. Please indicate where the dimer and the monomer are expected. What are the concentrations shown (please add to legend).

The calibration markers for the SEC column were on Figure 5E as filled triangles above the curves (labeled as 17 and 44 kDa). DIP-η IG1 runs ~30% smaller than its expected elution volume for a globular protein of same molecular weight, both for the monomer and dimer, likely due to transient non-specific interactions with gel filtration resin. While this kind of behavior is common, it does not affect our analysis of dimerization. Finally, we have added labels for protein concentrations. We calculate protein amounts in each peak not by the amounts loaded, but using the areas under each peak, to correct for any protein loss common to chromatography injections and runs.

Please explain why the magenta and mustard concentrations run so differently? Are these both monomers?

This comment, and others below, indicated to us that we missed an opportunity to explain in detail a feature of low-affinity, fast-kinetics complexes in our manuscript. When a protein is in a fast equilibrium of monomers and dimers (such as DIPs), one cannot observe separate monomer and dimer peaks in gel filtration, but a peak at one intermediate position determined by the ratio of monomers-to-dimers. This is conceptually identical to how NMR chemical shifts separate into peaks when a protein experiences slow-exchange kinetics but take single intermediate positions in fast-exchange kinetics. To our knowledge, this behavior was first rigorously demonstrated and simulated thirty years ago by Stevens, (1989). Further details of this behavior can be found in a protocol paper by his group (Wilton et al., 2004).

This phenomenon might be best simplified if one imagines a group of molecules traveling within a gel filtration column. Since the rates of P+P→P_2_ and P_2_→P+P conversions are fast and happening several times per second for a fast-kinetics dimer, any given P molecule can recycle through monomer and dimer many times within one gel filtration experiment: up to 1,800 times if *k*_ex_ = 1 s^-1^ and the elution time = 30 min. So, the average migration speed of the protein molecule will be more than the dimer’s but less than the monomer’s. The fast exchange rate will not allow monomeric and dimeric proteins to separate into peaks. In the case of slow kinetics (e.g., *k*_ex_ > 1 h^-1^), a protein is unlikely to convert between monomer to dimer during one chromatography run, and the two populations will separate into two peaks.

Stevens lab’s simulations describe how such chromatograms should appear in their 2004 paper. Panel A below shows actual data for a fast-exchange monomer/dimer with a *K*_D_ of 10 µM, and panel B shows expected chromatograms based on simulations. Our gel filtration results in Figure 5 and Figure 6 closely match their experimental and theoretical data.

We also would like to note that low-affinity/µM interactions almost always have fast kinetics, even though this is not theoretically required. In our case, our SPR results here and in Carrillo et al., (2015) show the fast kinetics.

In short, the mustard peak in Figure 5E belongs to protein molecules at a low concentration, which have spent most of their time on the SEC column in a fast-eluting monomeric state, and less time as a slow-eluting dimer. The magenta peak belongs to a sample 5x more concentrated, which has existed approximately same amount of time as a fast-eluting monomer and as a slow-eluting dimer. The peaks further to the left, at even higher concentrations, mostly existed as a dimer during the 1-hour long SEC experiment.

It is not clear in Figure 5 —figure supplement 1 how the curves in Figure 5E could be used to calculate the ratio of monomer to dimer, leading to an estimated KD of 10-40 µM. This is a key point used to argue that DIP homodimers form with somewhat less affinity than Dpr:DIP heteromers.

We have now corrected this oversight and expanded the methodology further in the manuscript. However, we would still like to emphasize that the evidence for heterodimers having higher affinity is not limited to these calculations.

Finally, we show in the manuscript that the Dpr-DIP heterodimer affinities may be underestimated (i.e. appear weaker) in SPR experiments due to competing homodimers. Therefore, it is more likely that we are underestimating heterophilic affinities, and the heterodimers are even more stronger than homodimers.

The legend of Figure 5 —figure supplement 1 states: "Dimer fraction was calculated by identifying monomeric and dimeric elution velocities and calculating monomer and dimer fractions for any given peak position." However, the trailing shoulders of the peaks in Figure 5E cannot be assumed to be dimer or monomer based on the elution position in the chromatogram alone because of protein loading issues. Large amounts of protein will have larger trailing peaks. So, the contents of the peaks must be validated by PAGE (native gels) to demonstrate dimer, monomer or a mixture.

We agree that when peaks trail it is problematic to assign an average elution volume, or migration speed, to the protein sample in the experiment. These “trailing shoulders” are expected (especially at intermediate concentrations), because the peak’s leading edge will start to separate from the peak and will migrate faster as the protein is more diluted and will lean further towards the monomeric state. This issue stem from the fact that in gel filtration chromatography the entire protein sample is not in equilibrium across the elution profile. Partly because of this, we report dissociation constants as a range, rather than precise numbers. Also, using an absorbance-weighted average value as the elution volume does not significantly change the apparent *K*_D_ values calculated from the binding isotherm.

The “trailing shoulders” is also a feature reported in Stevens’ simulations, explicitly mentioned in Wilton, Myatt and Stevens, (2004) further supporting our results.

Finally, the native PAGE experiment suggested by the reviewer will also not give multiple bands, but only an intermediate and probably fuzzy band, simply because the separation of PAGE bands happens on the timescale of minutes, while DIPs interconvert between monomer to dimer in the millisecond to second timescale.

Likewise, Figure 6D is surprising as well. If you have a monomer-dimer equilibrium you would expect two resolved peaks (e.g., the red curve for the dimer and the blue curve for the monomer). But a mixture should show two peaks/adjacent bumps (one for the dimer and one for the monomer) not something in the middle (green curve).

As explained above, in a fast-exchange monomer-dimer equilibrium, one should only observe one intermediate peak, and no separate peaks. Only in the slow-exchange regime, or when a kinetic trap can prevent quick conversion between monomer to dimer, multiple peaks can be observed. We have indeed worked with such proteins and published such data before (the cell-cycle control protein Mad2 can exist in multiple states, which are stable for hours to days, and monomers or dimers can be separated (Hara et al., 2015)). For such behavior, the kinetics of oligomerization and dissociation should be happening in timescales of at least minutes to hours.

Is there perhaps a typo for the green curve in Figure 6D which reads 59 μM? Please show the calibration markers for the SEC column in Figure 6D.

There is no typo. We need to test DIP-α and DIP-η homodimerization at [DIP] concentrations in the high μM and even millimolar range, as a result of their weak affinities compared to the heterodimers. Figure 6D shows gel filtration results at 7.8 µM, 59 µM and 1.13 millimolar concentrations. It appears that only at 1.1 mM, DIP-α is in ≥90% dimer. Such weak affinities (and even worse) are observed for well-established cell adhesion complexes; classical Cadherins homodimerize with *K*_D_s in the range of 14 µM to 127 µM (Vendome et al., 2014), which are still strong enough to create adherens junctions.

The calibration markers are shown in Figure 6D as triangles above the curves. Unlike our DIP-η IG1 construct (pI = 8.7), DIP-α IG1 (pI = 6.9) elutes as monomer and dimer at volumes expected for a globular protein of the same size.

For all SPR data please add the model used (presumably 1:1 interaction model) to each curve so the reader can see how well the data fits the model and thus how accurate the KD are that are calculated based on the data. Please state in the legend which interaction model was used.

The binding model fits to the SPR data for all Dpr-DIP complexes are in Figure 4A-D, and all raw data are in the supplemental figures. This was mentioned in the legend for Figure 4 but needed to be further stated in the main text. We have added wording to the main text and expanded the Figure 4 legend. Also, the figure supplements include tables for each *K*_D_ measurement, and the standard error of mean values associated with those fits.

SPR data was fit using the 1:1 interaction model, except in Figure 5—figure supplement 1D, which measures DIP-η homodimerization using SPR. For this dataset, we modified binding equations to account for homodimerization in solution, and we fit the data to the modified model in MATLAB, as mentioned in the original manuscript. We had failed to include the equations used for DIP-η homodimer fitting in the Methods section, which has been remedied in the revised manuscript.